# SPHERICAL MESSAGE PASSING FOR 3D MOLECULAR GRAPHS

**Yi Liu**[*], **Limei Wang**[*], **Meng Liu, Yuchao Lin, Xuan Zhang, Bora Oztekin & Shuiwang Ji**
Department of Computer Science & Engineering
Texas A&M University
College Station, TX 77843, USA
`{yiliu,limei,mengliu,kruskallin,xuan.zhang,bora,sji}@tamu.edu`

## ABSTRACT

We consider representation learning of 3D molecular graphs in which each atom is associated with a spatial position in 3D. This is an under-explored area of research, and a principled message passing framework is currently lacking. In this work, we conduct analyses in the spherical coordinate system (SCS) for the complete identification of 3D graph structures. Based on such observations, we propose the spherical message passing (SMP) as a novel and powerful scheme for 3D molecular learning. SMP dramatically reduces training complexity, enabling it to perform efficiently on large-scale molecules. In addition, SMP is capable of distinguishing almost all molecular structures, and the uncovered cases may not exist in practice. Based on meaningful physically-based representations of 3D information, we further propose the SphereNet for 3D molecular learning. Experimental results demonstrate that the use of meaningful 3D information in SphereNet leads to significant performance improvements in prediction tasks. Our results also demonstrate the advantages of SphereNet in terms of capability, efficiency, and scalability. Our code is publicly available as part of the DIG library (https://github.com/divelab/DIG).

## 1 INTRODUCTION

In many real-world studies, structured objects such as molecules are naturally modeled as graphs (Gori et al., 2005; Wu et al., 2018; Shervashidze et al., 2011; Fout et al., 2017; Liu et al., 2020; Wang et al., 2020). With the advances of deep learning, graph neural networks (GNNs) have been developed for learning from graph data (Kipf & Welling, 2017; Defferrard et al., 2016; Veličković et al., 2018; Zhang et al., 2018; Xu et al., 2019; Gao & Ji, 2019; Gao et al., 2018; 2020). Currently, the message passing scheme (Gilmer et al., 2017; Sanchez-Gonzalez et al., 2020; Vignac et al., 2020; Battaglia et al., 2018) is one of the commonly used architectures for realizing GNNs. In this work, we aim at developing a novel message passing method for 3D graphs. Generally, a 3D molecular graph contains 3D coordinates for each atom given in the Cartesian system along with the graph structure (Liu et al., 2019; Townshend et al., 2019; Axelrod & Gomez-Bombarelli, 2020). Different types of relative 3D information can be derived from 3D molecular graphs, and they can be important in molecular learning, such as bond lengths, angles between bonds (Schütt et al., 2017; Klicpera et al., 2020b).

We first investigate complete representations of 3D molecules. This requires the graph structure to be uniquely defined by relative 3D information. To this end, we conduct formal analyses in the spherical coordinate system (SCS) (Chen et al., 2019), and show that relative location of each atom in a 3D graph is uniquely determined by three geometries, including distance, angle, and torsion. However, such completeness needs to involve edge-based 2-hop information, leading to excessively high computational complexity. To circumvent the computational cost, we propose a novel message passing scheme, known as the spherical message passing (SMP), for fast and accurate 3D molecular learning. Our SMP is efficient and approximately complete in representing 3D molecules. First, we design a novel strategy to compute torsion, which only considers edge-based 1-hop information, thus substantially reducing training complexity. This enables the generalization of SMP to large-scale

---

[*]These authors contribute equally to this work.

molecules. In addition, we show that our SMP can distinguish almost all 3D graph structures. The uncovered cases seem rarely appear in practice. By naturally using relative 3D information and a novel torsion, SMP yields predictions that are invariant to translation and rotation of input graphs.

We apply the SMP to real-world molecular learning, where meaningful physical representations are needed. Geometries $(d, \theta, \varphi)$ specified by SMP are then physically represented by $\Psi(d, \theta, \varphi)$, which can be a solution to the Schrödinger equation, as described in Sec. 4. Based on this, we develop the spherical message passing neural networks, known as the SphereNet, for 3D molecular learning. We conduct experiments on various types of datasets including OC20, QM9, and MD17. Results show that, compared with baseline methods, SphereNet achieves the best performance without increasing the computing budget. Ablation study reveals contributions and necessity of different types of 3D information, including distance, angle, and torsion. Particularly, we compare with a complete message passing scheme that can distinguish all 3D graph structures but involves edge-based 2-hop information. Experimental results show that SphereNet achieves comparable performance but reduces running time by 4 times. This suggests the use of SphereNet in practice rather than the complete message passing scheme, whose computational complexity prevents its use on large molecules.

## 2 COMPLETE REPRESENTATIONS OF MOLECULES

Equivariant graph neural networks (EGNNs) represent one research area for 3D molecular graphs, as introduced in Sec. 5.1. These methods usually take coordinates in the Cartesian coordinate system (CCS) for all atoms as the raw input. Hence, all the network layers need to be carefully designed to be equivariant. The computing of some equivariant components is expensive, like spherical harmonics and Clebsh-Gordan coefficients (Thomas et al., 2018; Fuchs et al., 2020). In addition, the complicated SE(3) group representations may not be necessary for molecular learning where final representations are generally required to be invariant. In this work, we focus on the other category of methods that take relative position information purely as input to graph learning models. Relative 3D information could be distance or angle, which is inherently invariant to translation and rotation of input molecules. It is natural to consider such information in the spherical coordinate system (SCS). We start by investigating the structure identification of 3D molecules in the SCS. For any point in the SCS, its location is specified by a 3-tuple $(d, \theta, \varphi)$, where $d$, $\theta$, and $\varphi$ denote the radial distance, polar angle, and the azimuthal angle, respectively. When modeling 3D molecular graphs in the SCS, any atom $i$ can be the origin of a local SCS, and $d$, $\theta$, and $\varphi$ naturally become the bond length, the angle between bonds, and the torsion angle, respectively. Thus, the relative location of each neighboring atom of atom $i$ can be specified by the corresponding tuple $(d, \theta, \varphi)$. Similarly, the relative location of each atom in the 3D molecular graph can be determined, leading to the identified structure, which is naturally invariant to translation and rotation of the input graph. The SCS can be easily converted from the Cartesian coordinate system, thus, the tuple $(d, \theta, \varphi)$ can be easily obtained.

As in Fig. 1, we use the chemical structure of the hydrogen peroxide ($H_2O_2$) to show how $d$, $\theta$, and $\varphi$ are vital for the molecular structure identification. It is obvious that the structure is uniquely defined by the three bond lengths $d_1$, $d_2$, $d_3$, the two bond angles $\theta_1$, $\theta_2$, and the torsion angle $\varphi$. Note that the input may not contain all pairwise distances (all possible bond lengths). This is because the atomic connectivity is usually based on real chemical bonds

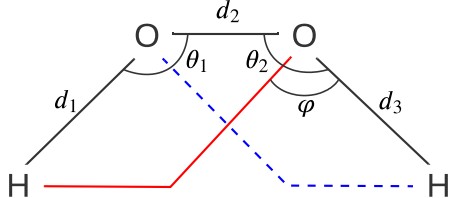

Figure 1: The chemical structure of the $H_2O_2$.

and cut-off distances. The cut-off distance is usually set as a hyperparameter. It is hard to guarantee that the cut-off is larger than any pairwise distance in a molecule. Hence, in this example, H-H bond length may not be considered as input if the cut-off is small. Setting a proper cut-off is even harder for other complicated and large molecules where a distance between two atoms could be large. In addition, considering all pairwise distances will cause severe redundancies, dramatically increasing the computational complexity. The model also easily gets confused by excessive noise, leading to unsatisfactory performance. From the perspective of completeness, using all pairwise distance is not capable of recognizing the chirality property. To overcome the above challenges, we use a combination of distance, angle, and torsion for rigorous design and accurate learning. Apparently, the two O-H bonds can rotate around the O-O bond without changing any of the bond lengths and bond angles. In this situation, however, the torsion angle $\varphi$ changes and the structure of the $H_2O_2$

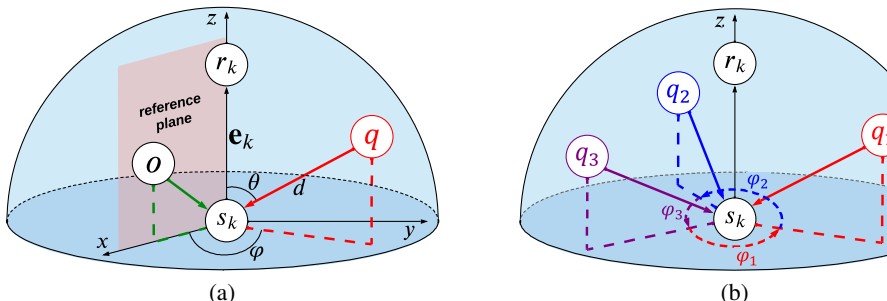

(a)                                                            (b)

Figure 2: (a). The message aggregation scheme for the spherical message passing. (b). An illustration for computing torsion angles in the spherical message passing architecture.

varies accordingly. The importance of torsion angle has also been demonstrated in related research domains. Garg et al. (2020) formally shows that the torsion along with the port numbering can improve the expressive power of GNNs in distinguishing geometric graph properties, such as girth and circumference, *etc*. Other studies (Ingraham et al., 2019; Simm et al., 2020) reveal that protein sequences and molecules can be accurately generated by considering the torsion in the given 3D structures. In this work, we propose SMP to systematically consider distance, angle, and torsion for approximately complete representation learning of 3D molecular graphs. Note that by using angle and torsion, SMP can easily recognize the chirality property.

## 3    SPHERICAL MESSAGE PASSING

### 3.1    MESSAGE PASSING SCHEME

Currently, the class of message passing neural networks (MPNNs) (Gilmer et al., 2017) are one of the most widely used architectures for GNNs. Based upon the completeness analyses in Sec. 2, we propose to perform message passing in the spherical coordinate system (SCS), resulting in a novel and efficient scheme known as spherical message passing (SMP). We show that message passing schemes used in existing methods, such as SchNet and DimeNet, are special cases of SMP.

We first formally define a 3D molecular graph, which is usually represented as a 4-tuple $G = (\mathbf{u}, V, E, P)$. The $\mathbf{u} \in \mathbb{R}^{d_{\mathbf{u}}}$ is a global feature vector for the molecular graph $G$. $V = \{\mathbf{v}_i\}_{i=1:n}$ is the set of atom features, where each $\mathbf{v}_i \in \mathbb{R}^{d_v}$ is the feature vector for the atom $i$. $E = \{(\mathbf{e}_k, r_k, s_k)\}_{k=1:m}$ is the set of edges, where each $\mathbf{e}_k \in \mathbb{R}^{d_e}$ is the feature vector, $r_k$ is the index of the receiver atom, and $s_k$ is the index of the sender atom for the edge $k$. $P = \{\mathbf{r}_h\}_{h=1:n}$ is the set of 3D Cartesian coordinates that contains 3D spatial information for each atom. In addition, we let $E_i = \{(\mathbf{e}_k, r_k, s_k)\}_{r_k=i,k=1:m}$ denote the set of edges that point to the atom $i$, and $\mathcal{N}_i$ denote the indices of incoming nodes of atom $i$. The outputs after a message passing process include the updated global feature vector $\mathbf{u}' \in \mathbb{R}^{d_{\mathbf{u}}}$, the updated atom features $V' = \{\mathbf{v}'_i\}_{i=1:n}$, and the updated edges $E' = \{(\mathbf{e}'_k, r_k, s_k)\}_{k=1:m}$.

An illustration of the message aggregation scheme for SMP is provided in Fig. 2 (a). Apparently, the embedding of the atom $r_k$ is obtained by aggregating each incoming message $\mathbf{e}_k$. The message $\mathbf{e}_k$ is updated based on $E_{s_k}$, the set of incoming messages pointing to the atom $s_k$. Let $q$ denote the sender atom of any message in $E_{s_k}$. Hence, we can define a local SCS, where $s_k$ serves as the origin, and the direction of the message $\mathbf{e}_k$ naturally serves as the $z$-axis. We define a neighboring atom $o$ of $s_k$ as the reference atom. Thus, the reference plane is formed by three atoms $s_k$, $r_k$, and $o$. For atom $q$, its location is uniquely defined by the tuple $(d, \theta, \varphi)$, as shown in Fig. 2 (a). Specifically, $d$ determines its distance to the atom $s_k$, $\theta$ specifies its direction to update the message $\mathbf{e}_k$. The torsion angle $\varphi$ is formed by the defined reference plane and the plane spanned by $s_k$, $r_k$, and $q$. Intuitively, as an advanced message passing architecture in spherical coordinates for 3D graphs, SMP specifies relative location for any neighboring atom $q$ by considering all the distance, angle, and torsion information, leading to more comprehensive representations for 3D molecular graphs.

Generally, the atom $s_k$ may have several neighboring atoms, which we denote as $q_1, ..., q_t$. It is easy to compute the corresponding bond lengths and bond angles for these $t$ atoms. The SMP computes torsion angles by projecting all the $t$ atoms to the plane that is perpendicular to $\mathbf{e}_k$ and intersect

with $s_k$. Then on this plane, the torsion angles are formed in a predefined direction, such as the anticlockwise direction. By doing this, any atom naturally becomes the reference atom for its next atom in the anticlockwise direction. Notably, the sum of these $t$ torsion angles is $2\pi$. A simplified case is illustrated in Fig. 2 (b). The atom $s_k$ has three neighboring atoms $q_1$, $q_2$, and $q_3$; $q_3$ is the reference atom for $q_1$, and they form $\varphi_1$; $q_1$ is the reference atom for $q_2$, and they form $\varphi_2$; similarly, $q_2$ is the reference atom for $q_3$, and they form $\varphi_3$. It is obvious that the sum of $\varphi_1$, $\varphi_2$, and $\varphi_3$ is $2\pi$. As the torsion is defined relatively, $q_1$ can be picked arbitrarily, which will not affect the output of the message passing scheme, as we perform summation when aggregating information to $s_k$ from its neighbors $q_1$, $q_2$, and $q_3$. Notably, by designing each atom to be the reference atom of the next one in the predefined direction, invariance is effectively achieved because reference atom is naturally relative. In addition, our method computes torsion within edge-based 1-hop neighborhood. Even though a torsion angle involves four atoms, our design avoids the number of torsion angles to be exponential, but makes it the same as the number of neighboring atoms. Hence, it is efficient and will not cause time or memory issues. Formally, the proposed SMP can be defined in the SCS as

$$\begin{aligned}
\mathbf{e}'_k &= \phi^e \left( \mathbf{e}_k, \mathbf{v}_{r_k}, \mathbf{v}_{s_k}, E_{s_k}, \rho^{p\to e} \left( \{\mathbf{r}_h\}_{h=r_k \cup s_k \cup \mathcal{N}_{s_k}} \right) \right), \\
\mathbf{v}'_i &= \phi^v \left( \mathbf{v}_i, \rho^{e\to v} \left( E'_i \right) \right), \mathbf{u}' = \phi^u \left( \mathbf{u}, \rho^{v\to u} \left( V' \right) \right),
\end{aligned}$$

(1)

where $\phi^e$, $\phi^v$, and $\phi^u$ are three information update functions on edges, atoms, and the whole graph, respectively. $\rho^{e\to v}$ and $\rho^{v\to u}$ aggregate information between different types of geometries. Especially, in SMP, the 3D information in $P$ is converted and incorporated to update each message $\mathbf{e}^k$. Hence, SMP employs another position aggregation function $\rho^{p\to e}$ for message update. Notably, the main difference between our SMP scheme defined in Eq. 1 and the GN framework in Battaglia et al. (2018) is the inclusion of 3D information $P$. In line with the research area described in Sec. 5.1.2, we focus on such 3D information and develop a systematic solution to incorporate it completely and efficiently. Detailed description of these functions is given in Appendix A.

## 3.2 Completeness Versus Efficiency

The identification criteria described in Sec. 2 can fully determine the structure of a 3D molecule, but involves edge-based 2-hop information. Hence, the computational complexity is as sizeable as $O(nk^3)$, where $n$ is the number of atoms, and $k$ denotes the average number of neighboring atoms for each center atom. Unfortunately, such design can hardly generalize to large molecular graphs. To this end, we propose SMP as an efficient and scalable scheme to realize message passing in SCS. Our SMP only involves edge-based 1-hop information, thus the

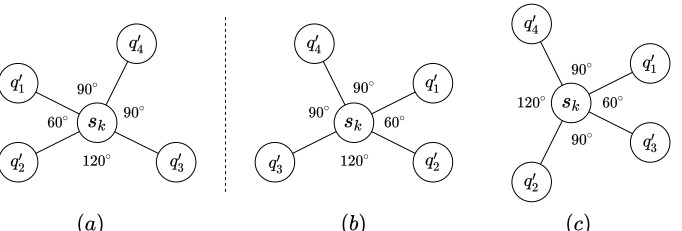

$(a)$      $(b)$      $(c)$

Figure 3: An illustration of cases that SMP can and cannot distinguish. All the neighboring nodes of $s_k$ are projected to the plate perpendicular to the message of interest. We assume all the distances and angles are fixed (the molecules can be more easily distinguished otherwise). Hence, all the angle shown are torsion angles and they are formed in the anticlockwise direction. (a) and (b) are chiral and SMP can distinguish them. This is because in (a), $q'_1(90°)$, $q'_2(60°)$, $q'_3(120°)$, $q'_4(90°)$; in (b), $q'_1(60°)$, $q'_2(120°)$, $q'_3(90°)$, $q'_4(90°)$. SMP cannot distinguish (b) and (c) but this scenario may not exist in nature. $\angle q'_1 s_k q'_2$ in (b) and $\angle q'_1 s_k q'_3$ in (c) usually are different as $q'_2$ and $q'_3$ are different atoms and the corresponding distances and angles are the same.

time complexity is reduced to $O(nk^2)$. This enables the application of SMP to large molecules, like the newly released OC20 data (Chanussot et al., 2020). We rigorously investigate the completeness of SMP and show that it can distinguish even complex geometric properties such as chirality, as indicated by Fig. 3 (a) and Fig. 3 (b). As SMP uses the last atom as the reference atom (like $q_2$ is the reference atom for $q_3$ in Fig. 2 (b)) in a predefined direction, the relative order between adjacent atoms is considered while the absolute order is neglected. Hence, SMP can not distinguish the two molecules illustrated by Fig. 3 (b) and Fig. 3 (c). However, this scenario may not exist in nature. This is also demonstrated in experiments that our SMP achieves comparable performance with the complete representations, while the latter induces huge time complexity and severe memory issues.

### 3.3 Relations with Prior Message Passing Methods

When developing message passing methods for 3D graphs, sphere message passing is an advanced scheme where the relative location of each atom is more specified. The development for 3D graphs with relative information is still in early stage. To our best knowledge, there exist several notable methods in literature, and they can be viewed as special cases of the SMP, as they capture partial 3D position information. For example, the SchNet and PhysNet consider distance, and the DimeNet encodes directional information. Formally, these methods can be perfectly fit the Spherical scheme defined in Eq. 1. We describe the details of these methods in Appendix C. Notably, compared with prior models, SMP provides a rigorous justification on its completeness with failure cases clearly described. Importantly, SMP is developed based on the identification analyses of 3D molecular graphs. Hence, it aims at learning complete data representations for 3D molecular graphs, rather than simply including extra 3D information (like angle or torsion).

## 4 SphereNet

The obtained 3-tuple $(d, \theta, \varphi)$ indicates the relative location of any atom in a 3D molecular graph. However, it cannot serve as the direct input to neural networks as it lacks meaningful representations. Essentially, molecules are quantum systems thus the representation design needs to follow physics laws. An important aspect is to choose appropriate basis functions that transform the 3-tuple $(d, \theta, \varphi)$ into physically-based representations. Several basis functions have been explored in Hu et al. (2021); Klicpera et al. (2020b), including MLP, Gaussian and sine functions, spherical Bessel basis, and spherical harmonics. Especially, spherical Bessel is shown to be the best basis for encoding distance, and spherical harmonics is the most appropriate one for encoding angle (Hu et al., 2021; Klicpera et al., 2020b). We denote the final representation as $\Psi(d, \theta, \varphi)$. Referring to theories in Griffiths & Schroeter (2018); Cohen et al. (2019); Klicpera et al. (2020b), one form of the representation can be expressed as $\Psi(d, \theta, \varphi) = j_\ell \left( \frac{\beta_{\ell n}}{c} d \right) Y_\ell^m(\theta, \varphi)$, where $j_\ell(\cdot)$ is a spherical Bessel function of order $\ell$, $Y_\ell^m$ is a spherical harmonic function of degree $m$ and order $\ell$, $c$ denotes the cutoff, $\beta_{\ell n}$ is the $n$-th root of the Bessel function of order $\ell$. We also have $\ell \in [0, \cdots, L-1]$, $m \in [-\ell, \cdots, \ell]$ and $n \in [1, \cdots, N]$. $L$ and $N$ denote the highest orders for the spherical harmonics and spherical Bessel functions, respectively. They are hyperparameters in experimental settings. In addition, we can derive two simplified representations $\Psi(d)$ and $\Psi(d, \theta)$ from $\Psi(d, \theta, \varphi)$.

Based upon the spherical message passing scheme described in Sec. 3 and physical representations, we build the SphereNet for molecular learning. Apparently, SphereNet can produce data representations that are both accurate and physically meaningful. By incorporating the position information in spherical coordinates, SphereNet also generates predictions invariant to translation and rotation of input molecules. Following the architecture design in the research line stated in Sec. 5.1.2, our network is composed of an input block, several interaction blocks, and an output block. For clear description, we assume the message $\mathbf{e}^k$ for the edge $k$ in Fig. 2 and Eq. (1) is the message for update. The update process and detailed architecture for the SphereNet are provided in Appendix B.

## 5 Related Work

### 5.1 Methods for 3D Molecular Graphs

#### 5.1.1 Equivariant Graph Neural Networks

One research line for 3D molecular graphs is equivariant graph neural networks (EGNNs) including tensor field networks (TFNs) (Thomas et al., 2018), SE(3)-transformers (Fuchs et al., 2020), PaiNN (Schütt et al., 2021), NequIP (Batzner et al., 2021), Noisy Nodes (Godwin et al., 2022), *etc.* The raw input of these methods usually contains the absolute information, such as coordinates in the Cartesian coordinate system. In intermediate layers, absolute information could be decomposed into partial absolute information and partial relative information as needed. A simple example is that a vector can be decomposed into its orientation (absolute) and length (relative) (Thomas et al., 2018; Schütt et al., 2021). Apparently, network components of these methods should be carefully designed to be equivariant. The preliminary work like TFNs were developed for 3D point clouds. However, it is demonstrated that for molecules whose downstream tasks usually require the systems

to be invariant, the complicated SE(3) group representations are not necessary but $S^2$ representations are enough (Klicpera et al., 2021). Moreover, their performance on molecular tasks is not satisfactory.

### 5.1.2 INVARIANT GRAPH NEURAL NETWORKS

Another category of methods purely take relative 3D information as input, such as distances between atoms, angles between bonds, angles between planes, *etc.* Hence, the network is naturally invariant. The development of these methods is in early stage, and existing studies focus on leveraging different geometries. The SchNet (Schütt et al., 2017) incorporates the distance information during the information aggregation stage by using continuous-filter convolutional layers. The PhysNet (Unke & Meuwly, 2019) integrates both the atom features and distance information in the proposed interaction block. The DimeNet (Klicpera et al., 2020b) is developed based on the PhysNet and moves a step forward by considering directional information in the interaction block. The GemNet (Klicpera et al., 2021) is proposed recently for universal molecular representations. The OrbNet (Qiao et al., 2020) combines distance information with the atomic orbital theory to design important SAAO features as inputs to GNNs. Generally, the use of 3D position information usually results in improved performance. However, existing methods simply include additional geometries such as distance and angle, and there lacks a rigorous justification on how different geometries contribute to the information aggregation process. We conduct formal analysis and show that all the distance, angle, and torsion are necessary for 3D molecular identification, based on which we propose SphereNet to generate more powerful molecular representations.

### 5.2 METHODS FOR OTHER OBJECTS MODELED AS GRAPHS

Besides molecules, many other data objects are also represented as graphs, such as 3D point clouds (Guo et al., 2020; Simonovsky & Komodakis, 2017; Shen et al., 2018; Landrieu & Simonovsky, 2018) and meshes (Bronstein et al., 2021; De Haan et al., 2020; Perraudin et al., 2019). When modeling 3D point clouds as 3D graphs, points are represented as nodes and connections between points are directed edges. Existing methods mainly capture distance information from local neighborhood in 3D space. In DGCNN (Wang et al., 2019b), a novel layer namely EdgeConv is proposed to aggregate distance-based edges features for node learning. In Landrieu & Boussaha (2019), neighborhood radius along with spatial orientation is incorporated in local point embedding. The work Wang et al. (2019a) proposes a graph attention convolution for 3D point clouds. Generally, these methods can be fit into our message passing scheme defined in Eq. 1. The work De Haan et al. (2020) is an exemplary study that formulates meshes as graphs with considering geometrical information. The used convolutional kernel depends on the angle between a predefined reference edge and any other edge projected to the tangent plane for each vertex. It focuses on the design of gauge equivariance rather than the learning of complete geometry information. In this work, we study the complete learning of 3D molecules and leave extensive studies on other data types as future work.

## 6 EXPERIMENTAL STUDIES

### 6.1 EXPERIMENTAL SETUP

We apply our SphereNet to three benchmark datasets, including Open Catalyst 2020 (OC20) (Chanussot et al., 2020), QM9 (Ramakrishnan et al., 2014), and MD17 (Chmiela et al., 2017). Baseline methods include PPGN (Maron et al., 2019), SchNet (Schütt et al., 2017), PhysNet (Unke & Meuwly, 2019), Cormorant (Anderson et al., 2019), PaiNN (Schütt et al., 2021), NequIP (Batzner et al., 2021), MGCN (Lu et al., 2019), DimeNet (Klicpera et al., 2020b), DimeNet++ (Klicpera et al., 2020a), GemNet (Klicpera et al., 2021), CGCNN (Xie & Grossman, 2018), and sGDML (Chmiela et al., 2018). Detailed configurations of all the models used in the following sections are provided in the supplementary material. Unless otherwise specified, for all the baseline methods, we report the results taken from the referred papers or provided by the original authors. For the SphereNet, all models are trained using the Adam optimizer (Kingma & Ba, 2014). The optimal hyperparameters are obtained by grid search. Network configurations and search space for all models are provided in Appendix D.

Table 1: Comparisons between SphereNet and other models on IS2RE in terms of energy MAE and the percentage of EwT of the ground truth energy. Results reported for models trained on the All training dataset. The best results are shown in bold.

| Model | Energy MAE [eV] ↓ | | | | | EwT ↑ | | | | |
|---|---|---|---|---|---|---|---|---|---|---|
| | ID | OOD Ads | OOD Cat | OOD Both | Average | ID | OOD Ads | OOD Cat | OOD Both | Average |
| CGCNN | 0.6203 | 0.7426 | 0.6001 | 0.6708 | 0.6585 | 3.36% | 2.11% | 3.53% | 2.29% | 2.82% |
| SchNet | 0.6465 | 0.7074 | 0.6475 | 0.6626 | 0.6660 | 2.96% | 2.22% | 3.03% | 2.38% | 2.65% |
| DimeNet++ | 0.5636 | 0.7127 | 0.5612 | 0.6492 | 0.6217 | 4.25% | 2.48% | 4.40% | 2.56% | 3.42% |
| GemNet-T | **0.5561** | 0.7342 | 0.5659 | 0.6964 | 0.6382 | 4.51% | 2.24% | 4.37% | 2.38% | 3.38% |
| **SphereNet** | 0.5632 | **0.6682** | **0.5590** | **0.6190** | **0.6024** | **4.56%** | **2.70%** | **4.59%** | **2.70%** | **3.64%** |

## 6.2   OC20

The Open Catalyst 2020 (OC20) dataset is a newly released large-scale dataset for catalyst discovery and optimization (Chanussot et al., 2020). It comprises millions of DFT relaxations across huge chemical structure space such that machine learning models can be fully trained. We focus on the IS2RE task in this work, and description of the data is provided in Appendix E. Results for CGCNN, SchNet, and DimeNet++ are provided in Chanussot et al. (2020). The original GemNet paper does not contain results on the OC20 dataset, We use the publicly available code from the OC Project website[1] to produce results for GemNet-T. We report evaluation results of fixed epochs for SphereNet. Following a setting in Chanussot et al. (2020), we use the direct approach and all the training data for training models. The used metrics are the energy MAE and the percentage of Energies within a Threshold (EwT) of the ground truth energy. Table 1 shows that the SphereNet achieves the best performance on 3 out of 4 splits and the average in terms of energy MAE. For EwT, SphereNet is the best on all the 4 splits. Specifically, it reduces the average energy MAE by 0.019, which is 3.10% of the second best model. In addition, it improves the average EwT from 3.42% to 3.64%, which is a large margin considering the inherently low EwT values.

Notably, ForceNet (Hu et al., 2021) and GemNet (Klicpera et al., 2021) are recently proposed for quantum system learning. A prominent advantage for ForceNet is its high efficiency and scalability to large molecules. ForceNet focuses on S2EF thus there are no original results for the IS2RE task. However, DimeNet++ is slightly better than ForceNet in terms of performance, and our SphereNet outperforms DimeNet++ significantly. GemNet has two variants GemNet-T and GemNet-Q. GemNet-T considers distance and angle information as input, and contains an effective architecture with novel network components, such as bilinear layers and scaling factors. We can see GemNet-T is similar as DimeNet++ in terms of performance. GemNet-Q is claimed to be able to capture universal representations of molecules. However, it considers edge-base 2-hop information and the time complexity is extremely high. It may not be configured properly on the large catalyst molecules.

## 6.3   QM9

We apply the SphereNet to the QM9 dataset, which is widely used for predicting various properties of molecules. It consists organic molecules composed of up to 9 heavy atoms. Thus, this test can examine the power of the SphereNet for similar quantum chemistry systems. The dataset is original split into three sets, where the training set contains 110,000, the validation set contains 10,000, and the test set contains 10,831 molecules. For energy-related properties, the training processes use the unit eV. All hyperparameters are tuned on the validation set and applied to the test set. We compare our SphereNet with baselines using mean absolute error (MAE) for each property and the overall mean standarized MAE (std. MAE) for all the 12 properties. The comparison results are summarized in Table 2. SphereNets achieves best performance on 5 properties and the second best performance on 3 properties. It also improves the overall mean std. MAE of the QM9 dataset from 0.98 to 0.91 and sets the new state of the art. Notably, the most recent method PaiNN uses the same data splits as SphereNet in terms of sample numbers. Its final performance is the average of three different runs on three random splits. We follow such settings and run SphereNet on four properties including $\epsilon_{HOMO}$, $\epsilon_{LUMO}$, $U_0$, and $\mu$. The corresponding results are $22.9 \pm 0.2$, $18.8 \pm 0.2$, $6.28 \pm 0.05$, and $0.0243 \pm 0.00$, respectively. It is obvious that these results are highly close to those in Table 2, thus, we can draw consistent conclusions.

---

[1]https://github.com/Open-Catalyst-Project/ocp

Table 2: Comparisons between SphereNet and other models in terms of MAE and the overall mean std. MAE on QM9. '-' denotes no results are reported in the referred papers for the corresponding properties. The best results are shown in bold and the second best results are shown with underlines.

| Property | Unit | PPGN | SchNet | PhysNet | Cormorant | MGCN | DimeNet | DimeNet++ | PaiNN | **SphereNet** |
|---|---|---|---|---|---|---|---|---|---|---|
| $\mu$ | D | 0.047 | 0.033 | 0.0529 | 0.13 | 0.0560 | 0.0286 | 0.0297 | **0.012** | 0.0245 |
| $\alpha$ | $a_0^3$ | 0.131 | 0.235 | 0.0615 | 0.092 | **0.0300** | 0.0469 | 0.0435 | 0.045 | 0.0449 |
| $\epsilon_{HOMO}$ | meV | 40.3 | 41 | 32.9 | 36 | 42.1 | 27.8 | 24.6 | 27.6 | **22.8** |
| $\epsilon_{LUMO}$ | meV | 32.7 | 34 | 24.7 | 36 | 57.4 | 19.7 | 19.5 | 20.4 | **18.9** |
| $\Delta\epsilon$ | meV | 60.0 | 63 | 42.5 | 60 | 64.2 | 34.8 | 32.6 | 45.7 | **31.1** |
| $\langle R^2 \rangle$ | $a_0^2$ | 0.592 | 0.073 | 0.765 | 0.673 | 0.110 | 0.331 | 0.331 | **0.066** | 0.268 |
| ZPVE | meV | 3.12 | 1.7 | 1.39 | 1.98 | **1.12** | 1.29 | 1.21 | 1.28 | **1.12** |
| $U_0$ | meV | 36.8 | 14 | 8.15 | 28 | 12.9 | 8.02 | 6.32 | **5.85** | 6.26 |
| $U$ | meV | 36.8 | 19 | 8.34 | - | 14.4 | 7.89 | 6.28 | **5.83** | 6.36 |
| $H$ | meV | 36.3 | 14 | 8.42 | - | 14.6 | 8.11 | 6.53 | **5.98** | 6.33 |
| $G$ | meV | 36.4 | 14 | 9.40 | - | 16.2 | 8.98 | 7.56 | **7.35** | 7.78 |
| $c_v$ | $\frac{cal}{mol\,K}$ | 0.055 | 0.033 | 0.0280 | 0.031 | 0.0380 | 0.0249 | 0.0230 | 0.024 | **0.0215** |
| std. MAE | % | 1.84 | 1.76 | 1.37 | 2.14 | 1.86 | 1.05 | 0.98 | 1.01 | **0.91** |

Table 3: Comparisons between SphereNets and other models in terms MAE of forces on MD17. WoFE indicates weight of force over energy in loss functions. Results of all baseline models are directed taken or adapted (if the unit varies) from the original papers, and SphereNet uses two WoFEs in line with the original papers of different baselines for fair comparisons. The best results are shown in bold and the second best results are shown with underlines.

| Molecule | WoFE = 100 | | | | WoFE = 1000 | | | |
|---|---|---|---|---|---|---|---|---|
| | sGDML | SchNet | DimeNet | **SphereNet** | NequIP | GemNet-T | GemNet-Q | SphereNet |
| Aspirin | 0.68 | 1.35 | 0.499 | **0.430** | 0.353 | 0.220 | 0.217 | 0.209 |
| Benzene | 0.20 | 0.31 | 0.187 | **0.178** | 0.186 | 0.145 | 0.145 | 0.147 |
| Ethanol | 0.33 | 0.39 | 0.230 | **0.208** | 0.204 | 0.086 | 0.088 | 0.091 |
| Malonaldehyde | 0.41 | 0.66 | 0.383 | **0.340** | 0.328 | 0.155 | 0.160 | 0.172 |
| Naphthalene | **0.11** | 0.58 | 0.215 | 0.178 | 0.105 | 0.055 | 0.051 | 0.048 |
| Salicylic acid | **0.28** | 0.85 | 0.374 | 0.360 | 0.242 | 0.127 | 0.125 | 0.113 |
| Toluene | **0.14** | 0.57 | 0.216 | 0.155 | 0.102 | 0.060 | 0.060 | 0.054 |
| Uracil | **0.24** | 0.56 | 0.301 | 0.267 | 0.173 | 0.097 | 0.104 | 0.106 |
| std. MAE | 1.11 | 2.38 | 1.10 | **0.97** | 0.79 | 0.45 | 0.45 | 0.44 |

## 6.4 MD17

The MD17 dataset is used to examine the expressive power of SphereNet for molecular dynamics simulations. Following the settings in Schütt et al. (2017); Klicpera et al. (2020b), we train a separate model for each molecule to predict atomic forces. We use 1000 samples for training, and each of the eight molecules has both the validation and test sets. Note that all the baseline models employ a joint loss of forces and conserved energy during training. In the original SchNet (Schütt et al., 2017) and DimeNet (Klicpera et al., 2020b) papers, the authors set the weight of force over energy (WoFE) to 100, while the NequIP (Batzner et al., 2021) and GemNet (Klicpera et al., 2021) papers use a weight of 1000. As the WoFE tends to affect the force prediction significantly, we perform SphereNet with both WoFE values for fair comparisons. PaiNN (Schütt et al., 2021) uses neither 100 nor 1000 as WoFE, so we do not compare with it on MD17. The results for forces are reported in Table 3. Note that for Benzene, all models are evaluated on Benzene17, thus, the result for sGDML is 0.20 rather than 0.06 (Benzene18). We can observe from the table that when WoFE is 100 for all models, SphereNet consistently outperforms SchNet and DimeNet by largin margins. Notably, sGDML is one of the original work that created the MD17 dataset with carefully-designed features. Compared with sGDML, SphereNet performs better on four and worse on the other four molecules, which is similar to DimeNet. One reason is sGDML incorporates molecular symmetries to boost precision, and different molecules have different symmetries. However, sGDML has poorer generalization

power to larger datasets without hand-engineered features. In addition, SphereNet achieves much better overall std. MAE than sGDML. When using the same WoFE that is 1000, SphereNet achieves similar results with GemNet in spite of that GemNet-T is of high complexity and contains carefully designed network components for performance boost.

## 6.5 COMPLETENESS VERSUS EFFICIENCY

The message passing scheme Q-MP in GemNet represents the edge-based 2-hop geometric message passing and can generate complete representations of 3D molecular graphs. We study the capability and efficiency of the proposed SMP by comparing with Q-MP. Specifically, We use the same backbone network for these two MP methods for fair comparisons. We extensively use two backbones, which are the SphereNet backbone introduced in Sec. 4 and the GemNet backbone proposed in Klicpera et al. (2021). We conduct experiments on MD17, reporting performance and average running time for all the 8 molecules per epoch using the same computing infrastructure (Nvidia GeForce RTX 2080 TI 11GB). Results are shown in Table 4, from which we can observe that on either backbone network, SMP achieves very similar results with Q-MP. However, the time cost is much less than SMP, which indicates it is much more efficient than Q-MP. Based on analyses in Sec. 3.2, SMP can distinguish almost all molecular structures, and the failure cases may not exist in nature. Hence, SMP performs similarly with Q-MP even though the latter is complete theoretically but not scalable in practice. We further compare the efficiency between SphereNet and other models in terms of parameters and time cost in Appendix F. SphereNet uses similar computing budget with others but achieves the best performance.

Table 4: Comparisons bewtween SMP and Q-MP on MD17 using two backbone networks.

| Molecule | SphereNet Backbone | | GemNet Backbone | |
|---|---|---|---|---|
| | SMP | Q-MP | SMP | Q-MP |
| Aspirin | 0.209 | 0.247 | 0.225 | 0.231 |
| Benzene | 0.147 | 0.153 | 0.144 | 0.149 |
| Ethanol | 0.091 | 0.102 | 0.089 | 0.083 |
| Malonaldehyde | 0.172 | 0.168 | 0.169 | 0.176 |
| Naphthalene | 0.048 | 0.057 | 0.063 | 0.062 |
| Salicylic acid | 0.113 | 0.125 | 0.111 | 0.114 |
| Toluene | 0.054 | 0.043 | 0.052 | 0.063 |
| Uracil | 0.106 | 0.106 | 0.098 | 0.113 |
| Time/Epoch (s) | 324 | 1270 | 295 | 1185 |

## 6.6 ABLATION STUDY

The proposed SMP considers all the distance, angle, and torsion, leading to more powerful data representations. We investigate contributions of different geometries to demonstrate the advances of our SMP. We remove torsion information from SMP which we denote as "SMP w/o $\varphi$"; we further remove angle information which we denote as "SMP w/o $(\theta, \varphi)$". The three message passing strategies are integrated to the same architecture with other network parts remaining the same. We evaluate these models on four molecules of MD17. Table 5 shows that SMP outperforms SMP w/o $\varphi$, and SMP w/o $\varphi$ outperforms "SMP w/o $(\theta, \varphi)$". These results demonstrate the effectiveness of angle and torsion information used in the SMP. The best performance of SMP further reveals that SMP represents an accurate scheme for 3D graphs. In addition, we provide visualization results for SphereNet filters in Appendix G to further show that all the distance, angle, and torsion information determine the structural semantics of filters.

Table 5: Comparisons among three message passing strategies on the same SphereNet architecture on the partial MD17 dataset.

| Molecule | SMP w/o $(\theta, \varphi)$ | SMP w/o $\varphi$ | SMP |
|---|---|---|---|
| Ethanol | 0.249 | 0.22 | 0.208 |
| Malonaldehyde | 0.550 | 0.360 | 0.340 |
| Naphthalene | 0.372 | 0.205 | 0.178 |
| Toluene | 0.446 | 0.182 | 0.155 |

## 7 CONCLUSIONS

3D information is important for molecules but there lacks a principled message passing framework to consider it. We first propose the spherical message passing as a unifying and efficient scheme that can achieve approximately complete representations of molecules without increasing computing budget. Based on SMP and meaningful physical representations, SphereNet is presented, and experiments on various types of datasets demonstrates its capability, efficiency, and scalibility.

## REPRODUCIBILITY STATEMENT

Detailed experimental setup is provided in Appendix D. Implementation hyper-parameters of SphereNet on all the three datasets OC20, QM9, and MD17 are given in Table 6, Table 7, and Table 8, respectively. Code is integrated in the DIG library (Liu et al., 2021) and available at https://github.com/divelab/DIG.

## ACKNOWLEDGMENTS

This work was supported in part by National Science Foundation grant IIS-1908198 and National Institutes of Health grant 1R21NS102828. We thank Hannes Stärk for his valuable suggestions and discussions when developing the methods.

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

# Spherical Message Passing for 3D Molecular Graphs: Appendix

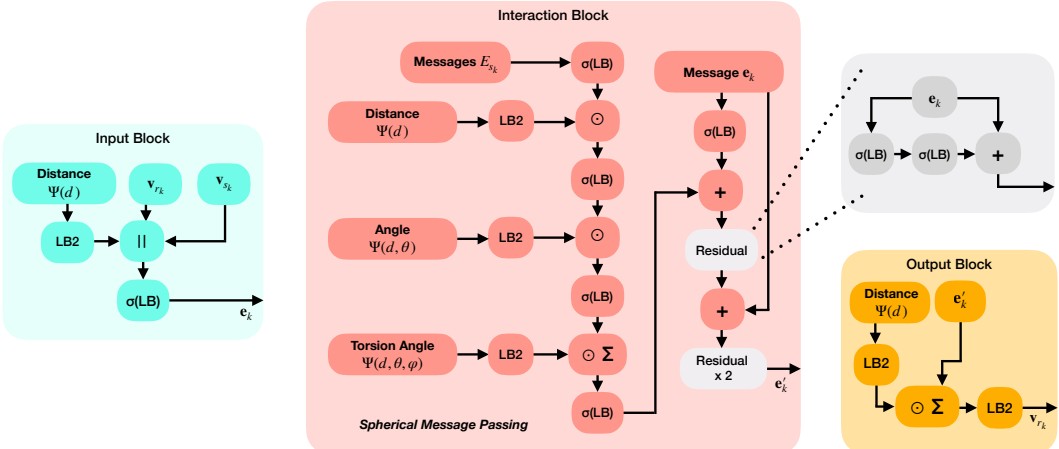

Figure 4: Architecture of SphereNet. LB2 denotes a linear block with two linear layers, $\sigma(\text{LB})$ denotes a linear layer followed by an activation function, $\parallel$ denotes concatenation, and $\odot$ denotes element-wise multiplication. Each LB2 aims at canceling bottlenecks by performing downprojection, followed by upprojection. Hence, it is related to three hyperparameters; these are, input embedding size, intermediate size, and output embedding size. Each linear block LB is related to hyperparameters including input embedding size and output embedding size. Description of each block is in Sec. B.

## A    UPDATE FUNCTIONS IN SMP

The function $\phi^e$ is applied to each edge $k$ and outputs the updated edge vector $\mathbf{e}'_k$. The indices of the input geometries to $\phi^e$ are illustrated in Fig. 5 (a). Correspondingly, the inputs include the old edge vector $\mathbf{e}_k$, the receiver node vector $\mathbf{v}_{r_k}$, the sender node vector $\mathbf{v}_{s_k}$, the set of edges $E_{s_k}$ that point to the node $s_k$, and the 3D position information for all the nodes connected by the edge $k$ and edges in $E_{s_k}$ with the index set as $r_k \cup s_k \cup \mathcal{N}_{s_k}$. The function $\rho^{p \to e}$ aggregates 3D information from these nodes to update the edge $k$. The function $\phi^v$ is used for per-node update and generates the new node vector $\mathbf{v}'_i$ for each node $i$. An illustration of the indices of the

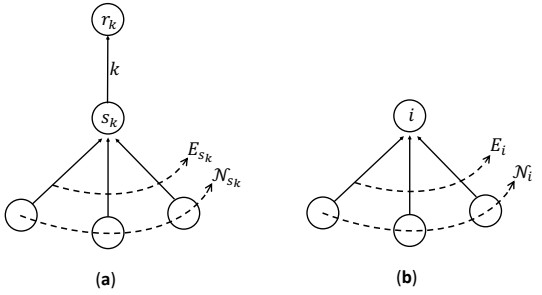

Figure 5: Illustrations of the functions $\phi^e$ (a) and $\phi^v$ (b).

inputs to $\phi^v$ is provided in Fig. 5 (b). The inputs include the old node vector $\mathbf{v}_i$, the set of edges $E'_i$ that point to the node $i$, and 3D information for all the related nodes (the index set is $i \cup \mathcal{N}_i$). The functions $\rho^{e \to v}$ is applied to aggregate the input edge features for updating the node $i$. The function $\phi^u$ is used to update the global graph feature, while the function $\rho^{v \to u}$ aggregates information from all the edge features.

The three information update functions $\phi^e$, $\phi^v$, and $\phi^u$ can be implemented in different ways, such as using neural networks and mathematical operations. In SMP, the 3D information in $P$ is converted and incorporated to update each message $\mathbf{e}^k$. Hence, SMP uses $\rho^{p \to e}$ compute position representations for edges. Note that as absolute Cartesian coordinates stored in $P$ are not invariant to translation and rotation, they are not used as immediate inputs to machine learning models. The position aggregation function can be flexibly adapted to generate invariant representations. For example, $\rho^{p \to e}$ in Eq. (1) can be adapted to a spherical Bessel basis function.

## B  INFORMATION UPDATE AND ARCHITECTURE OF SPHERENET

we assume the message $\mathbf{e}^k$ for the edge $k$ is the center message for update. The input block generates the initial message for the edge $k$, and takes only the distance representation $\Psi(d)$ as the input. Each interaction block updates the message for the edge $k$. The inputs include messages for all the neighboring edges, and all three representations, including $\Psi(d, \theta, \varphi)$, $\Psi(d, \theta)$, and $\Psi(d)$ based on the edge $k$ and its neighboring edges. The output block first takes both the distance representation and the current message for $k$ as inputs. Then the feature vector of the receiver atom for the edge $k$ (atom $r_k$ in Fig. 2 and Eq. (1)) is obtained by aggregating all the messages pointing to it, where other messages have a similar update process as $\mathbf{e}^k$.

Detailed architecture of SphereNet is provided in Fig. 4. Specifically, SphereNet is composed of an input block, followed by multiple interaction blocks and output blocks. For the purpose of simplicity, the architecture is explained by updating the receiver note $r_k$ of the message $\mathbf{e}_k$, as described in Eq. (1) and Sec. 4 in main paper.

**Input Block** aims at constructing initial message $\mathbf{e}_k$ for the edge $k$. Inputs include the distance representation $\Psi(d)$ for edge $k$, initial node embeddings $\mathbf{v}_{s_k}$, $\mathbf{v}_{r_k}$ for the sender node $s_k$, and the receiver node $r_k$. The distance information is encoded by using a LB2 block.

**Interaction Block** updates the message $\mathbf{e}_k$ with incorporating all the three physical representations. Input 3D information includes the distance embedding $\Psi(d)$, the angle $\Psi(d, \theta)$, and the torsion $\Psi(d, \theta, \varphi)$. The initial embedding sizes for them are $L$, $N \times L$, and $N^2 \times L$, respectively. Other inputs are old message $\mathbf{e}_k$ and the set of messages $E_{s_k}$ that point to the sender node $s_k$. Similar to the input block, each type of 3D information is encoded by using a block LB2. Note that each $\odot$ indicates the element-wise multiplication between the corresponding 3D information represented as a vector and each message in the set $E_{s_k}$. Thus, each neighboring message of $\mathbf{e}_k$ is gated by the encoded 3D information. The $\sum$ aggregates all the gated messages in $E_{s_k}$ to a vector, which is added to the transformation of the old message $\mathbf{e}_k$ as the updated message $\mathbf{e}'_k$. The transformation branch for old message $\mathbf{e}_k$ is composed of several nonlinear layers and residual blocks, as shown in Fig. 4.

**Output Block** aggregates all the incoming messages to update the feature for node $r_k$. Each incoming message has the same update process as $\mathbf{e}_k$ by interaction blocks. For the purpose of clear illustration, we use $\mathbf{e}'_k$ to represent each updated incoming message, which is further gated by the distance representation vector $\Psi(d)$.

## C  RELATIONS WITH PRIOR MESSAGE PASSING METHODS

Our SMP is a specific architecture for 3D graphs and is formally defined in Eq. (1). Especially, the message passing schemes used in some existing models can be viewed as special cases of SMP as they only encode partial 3D information. In this section, we clearly give the used functions as well as their inputs for each existing method. Basically, we describe how each method realizes Eq. (1).

### C.1  SCHNET, SCHÜTT ET AL. (2017)

In SchNet, the used aggregation function to encode 3D position information is $\rho^{p \to e}\left(\{\mathbf{r}_h\}_{h=r_k \cup s_k}\right) = \Psi\left(\|\mathbf{r}_{r_k} - \mathbf{r}_{s_k}\|\right)$, which converts the position information to an embedding of distance with radial basis functions. In addition to the $\rho^{p \to e}$ function, the $\phi^e$ function used is $\text{NN}\left(\text{NN}\left(\mathbf{v}_{r_k}\right) \odot \text{NN}\left(\Psi\left(\|\mathbf{r}_{r_k} - \mathbf{r}_{s_k}\|\right)\right)\right)$, where NN denotes a neural network and $\odot$ denotes the element-wise multiplication. The $\rho^{e \to v}$ function is $\sum_{\left(\mathbf{e}'_k, r_k, s_k\right) \in E'_i} \mathbf{e}'_k$. The $\phi^v$ function is $\mathbf{v}_i + \sum_{\left(\mathbf{e}'_k, r_k, s_k\right) \in E'_i} \mathbf{e}'_k$. The global feature $\mathbf{u}$ is updated based on the final node features $V^T$ and the

function is $\phi^u = \sum_{i=1:n} \mathrm{NN}\left(\mathbf{v}_i^T\right)$. Formally, the update process is expressed as

$$
\begin{aligned}
\mathbf{e}_k' =& \phi^e\left(\mathbf{v}_{r_k}, \rho^{p\to e}\left(\{\mathbf{r}_h\}_{h=r_k\cup s_k}\right)\right) \\
=& \phi^e\left(\mathbf{v}_{r_k}, \Psi\left(\|\mathbf{r}_{s_k} - \mathbf{r}_{r_k}\|\right)\right) \\
=& \mathrm{NN}\left(\mathrm{NN}\left(\mathbf{v}_{r_k}\right)\odot\mathrm{NN}\left(\Psi\left(\|\mathbf{r}_{r_k} - \mathbf{r}_{s_k}\|\right)\right)\right), \\
\mathbf{v}_i' =& \phi^v\left(\mathbf{v}_i, \rho^{e\to v}\left(E_i'\right)\right) \\
=& \phi^v\left(\mathbf{v}_i, \sum_{\left(\mathbf{e}_k', r_k, s_k\right)\in E_i'} \mathbf{e}_k'\right) \\
=& \mathbf{v}_i + \sum_{\left(\mathbf{e}_k', r_k, s_k\right)\in E_i'} \mathbf{e}_k', \\
\mathbf{u} =& \phi^u\left(\rho^{v\to u}\left(V^T\right)\right) \\
=& \sum_{i=1:n} \mathrm{NN}\left(\mathbf{v}_i^T\right).
\end{aligned}
\tag{2}
$$

## C.2 PHYSNET, UNKE & MEUWLY (2019)

PhysNet uses distance between atoms as an important feature and proposes more powerful neural networks for chemical applications. The position aggregation function is $\rho^{p\to e}\left(\{\mathbf{r}_h\}_{h=r_k\cup s_k}\right) = \Psi\left(\|\mathbf{r}_{r_k} - \mathbf{r}_{s_k}\|\right)$, where $\Psi$ is any radial basis function with a smooth cutoff. For the information update functions, the $\phi^e$ function is $\sigma\left(\mathbf{W}_1\right)\sigma\left(\mathbf{v}_{s_k}\right)\odot\mathbf{W}_2\Psi\left(\|\mathbf{r}_{r_k} - \mathbf{r}_{s_k}\|\right)$, the $\phi^v$ function is $\mathrm{NN}\left(\mathbf{W}_3\odot\mathbf{v}_i + \mathrm{NN}\left(\sigma\left(\mathbf{W}_4\right)\sigma\left(\mathbf{v}_i\right) + \sum_{\left(\mathbf{e}_k', r_k, s_k\right)\in E_i'}\mathbf{e}_k'\right)\right)$ and the $\phi^u$ function is $\mathbf{u} + \sum_{i=1:n}\mathrm{NN}\left(\mathbf{v}_i'\right)$. Here NN denotes a neural network, $\mathbf{W}_1, \mathbf{W}_2, \mathbf{W}_3, \mathbf{W}_4$ are learnable weight matrices, $\sigma$ is an activate function, and $\odot$ denotes the element-wise multiplication. PhysNet is expressed as

$$
\begin{aligned}
\mathbf{e}_k' =& \phi^e\left(\mathbf{v}_{s_k}, \rho^{p\to e}\left(\{\mathbf{r}_h\}_{h=r_k\cup s_k}\right)\right) \\
=& \phi^e\left(\mathbf{v}_{s_k}, \Psi\left(\|\mathbf{r}_{r_k} - \mathbf{r}_{s_k}\|\right)\right) \\
=& \sigma\left(\mathbf{W}_1\right)\sigma\left(\mathbf{v}_{s_k}\right)\odot\mathbf{W}_2\Psi\left(\|\mathbf{r}_{r_k} - \mathbf{r}_{s_k}\|\right), \\
\mathbf{v}_i' =& \phi^v\left(\mathbf{v}_i, \rho^{e\to v}\left(E_i'\right)\right) \\
=& \phi^v\left(\mathbf{v}_i, \sum_{\left(\mathbf{e}_k', r_k, s_k\right)\in E_i'} \mathbf{e}_k'\right) \\
=& \mathrm{NN}\left(\mathbf{W}_3\odot\mathbf{v}_i + \mathrm{NN}\left(\sigma\left(\mathbf{W}_4\right)\sigma\left(\mathbf{v}_i\right) + \sum_{\left(\mathbf{e}_k', r_k, s_k\right)\in E_i'}\mathbf{e}_k'\right)\right), \\
\mathbf{u}' =& \phi^u\left(\rho^{v\to u}\left(V'\right), \mathbf{u}\right) \\
=& \mathbf{u} + \sum_{i=1:n}\mathrm{NN}\left(\mathbf{v}_i'\right).
\end{aligned}
\tag{3}
$$

## C.3 DIMENET, KLICPERA ET AL. (2020B)

DimeNet explicitly considers distances between atoms and directions of directed edges. The aggregation functions on the position information is $\rho^{p\to e} = \left(\Psi\left(d\right)\|\Psi\left(d,\theta\right)\right)$, where $\|$ denotes concatenation, $\Psi\left(d\right)$ and $\Psi\left(d,\theta\right)$ are the same basis functions used in SphereNet as introduced in Sec. 4. Specifically, $\Psi\left(d\right)$ denotes the representation of the distance based on spherical Bessel function, and $\Psi\left(d,\theta\right)$ denotes the representation of distance and angle based on spherical Bessel function and spherical harmonics. For other functions, the $\phi^e$ function used is $\mathbf{e}_k' = \left(\mathbf{e}_{k,1}'\|\mathbf{e}_{k,2}'\right)$ with

$$\mathbf{e}_{k,1}' = \mathrm{NN}\left(\mathbf{e}_{k,1} + \mathrm{NN}\left(\sigma\mathbf{W}_1\mathbf{e}_{k,1} + \sum_{\left(\mathbf{e}_j, r_j, s_j\right)\in E_{s_k}} \mathbf{W}_2\Psi\left(d_j, \theta_{jk}\right)\left(\mathbf{W}_3\Psi\left(d_j\right)\odot\sigma\mathbf{W}_4\mathbf{e}_{j,1}\right)\right)\right)$$

and $\mathbf{e}'_{k,2} = \mathbf{W}_5 \Psi (d_j) \odot \mathbf{e}'_{k,1}$, where NN denotes a neural network, $\mathbf{W}_1, \mathbf{W}_2, \mathbf{W}_3, \mathbf{W}_4, \mathbf{W}_5$ are different weight matrices, and $\sigma$ is an activation function. The $\rho^{e \to v}$ function is $\sum_{(\mathbf{e}'_k, r_k, s_k) \in E'_i} \mathbf{e}'_{k,2}$ and the $\phi^v$ is NN $\left( \sum_{(\mathbf{e}'_k, r_k, s_k) \in E'_i} \mathbf{e}'_{k,2} \right)$. The $\rho^{v \to u}$ is $\sum_{i=1:n} \mathbf{v}'_i$ and the $\phi^u$ is $\mathbf{u} + \sum_{i=1:n} \mathbf{v}'_i$. Note that $\rho^{p \to v}, \rho^{p \to u}, \rho^{e \to u}$ functions are not required in DimeNet. The whole model is expressed as

$$
\begin{aligned}
\mathbf{e}_k &= (\mathbf{e}_{k,1} \| \mathbf{e}_{k,2}), \\
\rho^{p \to e} &= (\Psi (d) \| \Psi (d, \theta)), \\
\mathbf{e}'_{k,1} &= \phi^e \left( \mathbf{e}_k, E_{s_k}, \rho^{p \to e} \left( \{\mathbf{r}_h\}_{h = r_k \cup s_k \cup \mathcal{N}_{s_k}} \right) \right) \\
&= \text{NN} \left( \mathbf{e}_{k,1} + \text{NN} \left( \sigma \mathbf{W}_1 \mathbf{e}_{k,1} + \sum_{(\mathbf{e}_j, r_j, s_j) \in E_{s_k}} \mathbf{W}_2 \Psi (d_j, \theta_{jk}) (\mathbf{W}_3 \Psi (d_j) \odot \sigma \mathbf{W}_4 \mathbf{e}_{j,1}) \right) \right), \\
\mathbf{e}'_{k,2} &= \mathbf{W}_5 \Psi (d_j) \odot \mathbf{e}'_{k,1}, \\
\mathbf{v}'_i &= \phi^v (\rho^{e \to v} (E'_i)) \\
&= \text{NN} \left( \sum_{(\mathbf{e}'_k, r_k, s_k) \in E'_i} \mathbf{e}'_{k,2} \right), \\
\mathbf{u}' &= \phi^u (\mathbf{u}, \rho^{v \to u} (V')) \\
&= \mathbf{u} + \sum_{i=1:n} \mathbf{v}'_i.
\end{aligned}
\tag{4}
$$

## D  EXPERIMENTAL SETUP

For all the models used in three datasets, we set input embedding size = 256 and output embedding size = 64 for both LB2 and LB blocks. For each separate model, we first perform warmup on initial learning rate. Then two learning rate strategies, including ReduceLROnPlateau and StepLR, are used for training. For StepLR, the learning rate is decayed by the decay ratio every fixed epochs represented as step size. We do not use weight decay or dropout for all models. Some hyperparameters are fixed values, and some are tuned by grid search. Values/search space of hyperparameters for OC20, QM9, and MD17 are provided in Table 6, Table 7, and Table 8, respectively. Optimized hyperparameters are tuned on validation sets and applied to test sets for QM9 and MD17. For OC20, optimized hyperparameters are obtained on the ID split within max epochs, and then applied to the other three splits. Pytorch is used to implement all methods. For QM9 and MD17 datasets, all models are trained using one NVIDIA GeForce RTX 2080 Ti 11GB GPU. For the OC20 dataset, all models are trained using four NVIDIA RTX A6000 48GB GPUs.

## E  OC20 DATA DESCRIPTION

There exist three tasks including S2EF, IS2RS, and IS2RE. In this work, we focus on IS2RE that predicts structure's energy in the relaxed state. It is the most common task in catalysis as relaxed energies usually influence the catalyst activity. The dataset for IS2RE is originally split into training/validation/test sets. There are 460,318 structures in the training dataset in total. The test label is not publicly available. Performance is evaluated on the validation set, which has four splits including In Domain (ID), Out of Domain Adsorbates (OOD Ads), Out of Domain catalysts (OOD cat), and Out of Domain Adsorbates and catalysts (OOD Both), where numbers of structures are 24,943, 24,961, 24,963, 24,987, respectively. The average number of atoms per structure is 77.75.

## F  EFFICIENCY STUDY OF SPHERENET

We study the efficiency of SphereNet by comparing with other models regarding number of parameters and time cost per epoch using the same computing infrastructure (Nvidia GeForce RTX 2080 TI

Table 6: Values/search space for hyperparameters on OC20.

| Hyperparameters | Values/search space |
|---|---|
| Interaction block - distance LB2 intermediate size | 8 |
| Interaction block - angle LB2 intermediate size | 8 |
| Interaction block - torsion LB2 intermediate size | 8 |
| # of interaction blocks | 3, 4 |
| # of RBFs $N$ | 6 |
| # of spherical harmonics $L$ | 3, 5, 7 |
| Cutoff distance | 5, 6 |
| Batch size | 16, 32 |
| Initial learning rate | 1e-4, 5e-4, 1e-3 |
| Learning rate strategy | ReduceLROnPlateau, StepLR |
| Learning rate decay ratio (for StepLR) | 0.4, 0.5, 0.6 |
| Learning rate milestones (for StepLR) | 4, 7, 10, 12, 14 |
| Learning rate warmup epochs | 2 |
| Learning rate warmup factor | 0.2 |
| Max # of Epochs | 20 |

Table 7: Values/search space for hyperparameters on QM9.

| Hyperparameters | Values/search space |
|---|---|
| Interaction block - distance LB2 intermediate size | 4, 8, 16 |
| Interaction block - angle LB2 intermediate size | 4, 8, 16 |
| Interaction block - torsion LB2 intermediate size | 4, 8, 16 |
| # of interaction blocks | 3, 4, 5 |
| # of RBFs $N$ | 6 |
| # of spherical harmonics $L$ | 3, 5, 7 |
| Cutoff distance | 4, 5, 6 |
| Batch size | 32, 64 |
| Initial learning rate | 1e-4, 5e-4, 1e-3 |
| Learning rate strategy | StepLR |
| Learning rate decay ratio | 0.4, 0.5, 0.6 |
| Learning rate step size | 50, 100, 150 |
| Max # of Epochs | 500, 1000 |

11GB). Experiments are conducted on the property $U_0$ of QM9 and results are shown in Table 9. It is obvious that SphereNet uses similar computational resources as DimeNet++ and GemNet-T, and is much more efficient than DimeNet. The main reason could be we develop an efficient way to compute torsion, as introduced in Sec. 3 and Fig. 2 (b). Moreover, GemNet-Q cannot run on QM9 using the infrastructure as mentioned above.

Table 8: Values/search space for hyperparameters on MD17.

| Hyperparameters | Values/search space |
|---|---|
| Interaction block - distance LB2 intermediate size | 4, 8, 16 |
| Interaction block - angle LB2 intermediate size | 4, 8, 16 |
| Interaction block - torsion LB2 intermediate size | 4, 8, 16 |
| # of interaction blocks | 2, 3, 4, 5 |
| # of RBFs $N$ | 6 |
| # of spherical harmonics $L$ | 3, 5, 7 |
| Cutoff distance | 4, 5, 6 |
| Batch size | 1, 2, 4, 16, 32 |
| Initial learning rate | 1e-4, 5e-4, 1e-3 |
| Learning rate strategy | StepLR |
| Learning rate decay ratio | 0.4, 0.5, 0.6 |
| Learning rate step size | 50, 100, 200 |
| Max # of Epochs | 500, 1000, 2000 |

Table 9: Efficiency comparisons between SphereNet and other models in terms of number of parameters and time cost per epoch using the same infrastructure.

|  | SchNet | DimeNet | DimeNet++ | GemNet-T | SphereNet |
|---|---|---|---|---|---|
| #Param. | 185,153 | 2100,070 | 1887,110 | 2040,194 | 1898,566 |
| Time (s) | 100 | 840 | 240 | 290 | 340 |

## G    SPHERENET FILTER VISUALIZATION

We visualize SphereNet filters from a learned SphereNet model. Specifically, we port learned weights for the block LB2 after the torsion embedding $\Psi(d, \theta, \varphi)$ in Fig. 4. For each location represented by a tuple $(d, \theta, \varphi)$, the initial embedding size is $N^2 \times L$. The computation for the above LB2 is $\mathbf{W}_1 (\mathbf{W}_2 \Psi(d, \theta, \varphi))$, which results in the new embedding size of 64 for each location $(d, \theta, \varphi)$. We then perform sampling on locations in 3D space for visualizing weights as SphereNet filters. The visualization results are provided in Fig. 6. We set sampling rate in the torsion direction to be $\pi/4$, thus, there are eight samples in the torsion direction. There are totally 64 elements for each location, and we randomly pick 6 elements. Apparently, among the distance, angle and torsion, considering any one when fixing the other two, the structural value of filters will be different when the one of interest changes. It essentially shows that all the distance, angle, and torsion information determine the structural semantics of filters. This further demonstrates that SMP enables the learning of different 3D information for improved representations.

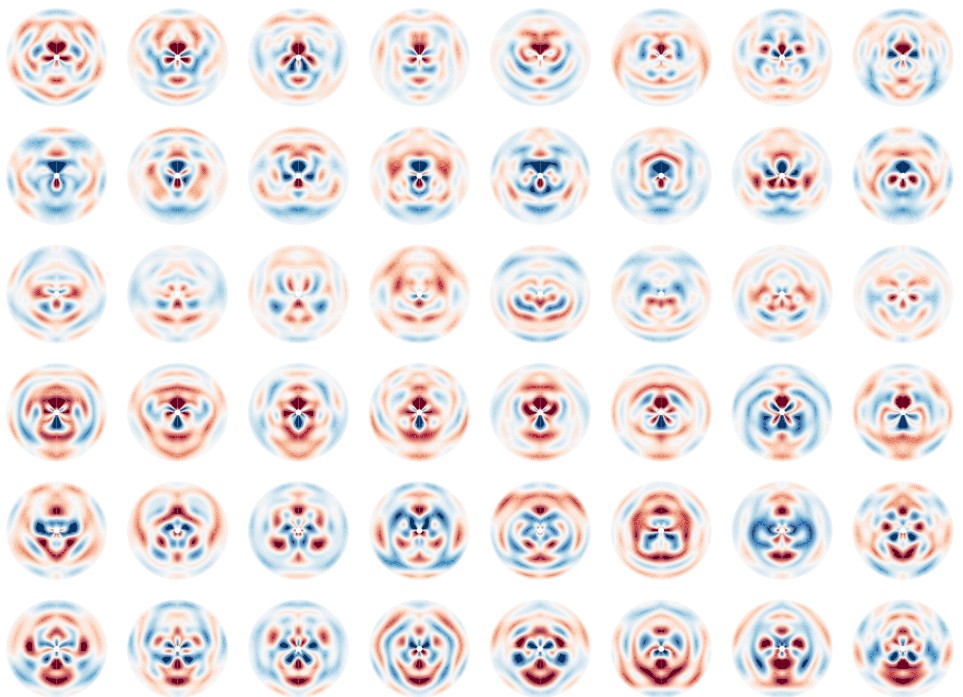

Figure 6: Visualization of six SphereNet filters. Each row corresponds to a filter with torsion angles $0$, $\pi/4$, $\pi/2$, $3\pi/4$, $\pi$, $5\pi/4$, $3\pi/2$, and $7\pi/4$ from left to right.

