# OpenReview forum: "Spherical Message Passing for 3D Molecular Graphs"
_ICLR.cc/2022/Conference — ICLR 2022 Poster_

### Official Review · Reviewer_6Y5X · 2021-10-28

**Correctness:** 4
**Technical Novelty And Significance:** 3
**Empirical Novelty And Significance:** 3
**Recommendation:** 8
**Confidence:** 3

**Main Review:**

As a premise, I am not an expert of molecular studies; in my review, I will focus on the technical and geometrical perspective, but I cannot assess the relevance of the method\results for the biology community.

==STRENGTHS==

ANALYSIS & REPLICABILITY
The experiments investigate different aspects of the method, providing insights on performance vs efficiency, ablation of used schema, and all the details of experimental setups and implementation. I think the paper can be replicable, and the claims are motivated.

NOVELTY
To my best knowledge, the idea is novel. While message passing methods on graphs exploit only the connectivity, this work shows an interesting method to include the embedding information in the case of geometrical graphs. On this point, however, I would suggest a better contextualization. Geometrical graphs are well explained in this book [1], and other works propose methods to exploit such geometrical prior (e.g., [2]). I would consider this in the related work section.

==WEAKNESSES==

COMPARISONS
The majority of the results are taken from other papers without re-running the experiments. I think this could be ok, but it is not completely clear if the split of training\val\test is preserved across the works (i.e., they are given by dataset) or if each method uses its own. I think it would be good to provide an average across multiple runs or reimplement at least the more recent competitors in this second case.

APPLICABILITY
The method explicitly addresses a specific domain, and I am not sure that it could be exported to other ones, e.g., meshes. How would it perform on geometrical graphs with thousands of nodes? The paper left such question for future works, but I think at least a computational timing on different domains would be useful to understand which kind of interest\impact it could receive in ICLR audience. I am not from the biological field, and I do not get the dimensions of the considered graphs (but being molecules, I assume that the order of magnitude is tens of nodes, is it correct?)

[1]: Geometric Deep Learning: Grids, Groups, Graphs, Geodesics, and Gauges; Bronstein et al., 2021
[2]: Gauge Equivariant Mesh CNNs Anisotropic convolutions on geometric graphs; Haan et al., 2020


**Summary Of The Paper:**

The paper proposes a message passing scheme using spherical coordinates. It is tested on three datasets of 3D moleclular graphs. The paper offer an in depth analysis of different aspects, with an extensive experimentation of the method.

**Summary Of The Review:**

While I am not sure that the method can be applied in other contexts, I think the idea is valuable, and the experiments are convincing, and I am prone to accept the paper.

---

> ### Author Response · Authors · 2021-11-15
> **Related work revised; experiments on meshes done; re-ran experiments over multiple runs when comparing with a more recent work. (2nd part)**
>
> > Comparisons using the same data splits and re-running experiments
>
> For OC20 and MD17, the baselines and SphereNet use the same data splits. it's a little tricky on QM9.
> The results of old baselines PPGN, PyhsNet, Cormorant, and MGCN are taken from recent work DimeNet [2] and DimeNet++ [3]. The results of SchNet are taken from the original paper [3] and are consistent in [2] and [3]. These baselines use the same training data and are a little different in val and test splits. Some use 110000/10000/10831 for training/val/test splits, while some use half val and half test. Given the difference is minimal and their results are much worse than SphereNet, we don't reimplement these methods. SphereNet uses exactly the same data splits as DimeNet and DimeNet++. As you suggest, we re-run the experiments when comparing with the most recent work PiaNN [5]. **We also revised Sec 6.3 of the paper for clarification**.
>
> PiaNN is the most recent work published in ICML 2021, and also achieves very good performance. The official code is not open-sourced yet, but the data split and training strategy are clear in the paper- 110k for training, 10k for val, and the rest for test; results are averaged over 3 random splits. Hence, we re-ran SphereNet over 3 random splits on 4 properties also. We picked 4 properties that SphereNet is better on 2 and worse on the other two. Results are
>
> | **Property**| *PaiNN* |*SphereNet* |*SphereNet (re-run)*|
> | ----------- | ----------- | ----------- | ----------- |
> | $\epsilon_\text{HOMO}$|$27.6$|$22.8$|$22.9\pm0.2$|
> | $\epsilon_\text{LUMO}$|$20.4$|$18.9$|$18.8\pm0.2$ |
> | $U_0$|$5.85$|$6.26$|$6.28\pm0.05$|
> | $\mu$|$0.012$|$0.0245$|$0.0243\pm0.00$|
>
> We can see from the results that SphereNet is similar to SphereNet (re-run).
> They both use the 110000/10000/10831 for training/val/test splits, but SphereNet (re-run) follows PaiNN that uses 3 random splits then performs the average. Hence, the effects of different random seeds are minimal, which may be due to the QM9 dataset being stable. Hence, we provide fair comparisons with the most recent work PaiNN and can draw the consistent conclusion that SphereNet is the most powerful.
>
> > Reference
>
> [1] Gauge Equivariant Mesh CNNs Anisotropic convolutions on geometric graphs
>
> [2] Directional message passing for molecular graphs
>
> [3] Fast and Uncertainty-Aware Directional Message Passing for Non-Equilibrium Molecules
>
> [4] SchNet: A continuous-filter convolutional neural network for modeling quantum interactions
>
> [5] Equivariant message passing for the prediction of tensorial properties and molecular spectra

---

> > ### Comment · Reviewer_6Y5X · 2021-11-18
> > **After authors' reply**
> >
> > I would acknowledge the authors' availability to answer the raised concerns, also by performing extra experiments. I have also carefully read the other reviews and the authors rebuttal.
> >
> > From my side, I am now more convinced by the method's effectiveness and its applicability to other domains. Reviewer epnb raised doubts about novelty, while the first strength point highlights that introducing torsion is actually a novelty. I would like to know if the rebuttal changed something, or there are any further comments.
> >
> > At the present state, I am prone to accept the paper, and I raise my score.

---

> ### Author Response · Authors · 2021-11-15
> **Related work revised; experiments on meshes done; re-ran experiments over multiple runs when comparing with a more recent work. (1st part)**
>
> Thanks a lot for your valuable comments! we have revised the paper and provided point-to-point responses as below.
>
> > A better contextualization and extensive related work
>
> Thanks for your suggestion! even though we study molecular learning (itself is an important and hot ML area) in this work, we totally agree that it's better to have a more comprehensive contextualization and to discuss if our method can be potentially applied to other domains. **We have revised the Sec. 5.2 of the paper and conducted more experiments** (which will be detailed in the next point).
>
> In the current version, we include two additional research areas: 3D point clouds and meshes. Many important methods for modeling 3D point clouds as graphs can be fit into our framework defined in Eq.1. This is to say, our idea of considering 3D information completely and efficiently is promising to be applied to this area. For meshes, we examine the literature and find only a part of the studies are graph based methods. The work [1] distinguishes from other graph based methods that it considers geometrical information (geometrical graphs), which is related to our paper. For each vertex, It basically picks an edge as reference edge, then computes angles of other edges with the reference edge when projected to the tangent plane. Now each edge's kernel value depends on the angle information w.r.t gauge. This is how it incorporates geometrical information---angle information. Our method studies the key problem---what geometrical information to consider and how to incorporate it for complete graph representations. **Hence, it's highly promising that our method can be used in this area in the future if completeness without information loss becomes a bottleneck in this area**. Another interesting observation is in [1], changed gauge would lead to changed output, hence, gauge equivariance is necessary. This is also related to equivariant GNNs in molecules, as stated in Sec 5.1.1.
>
> > The application of SphereNet on large graphs like meshes with thousands of nodes
>
> We demonstrated the power of our method on large molecules, as in Sec 6.2. A molecular graph (catalyst material) could contain as many as 200 nodes, which is already huge considering that real molecules in nature are small. As our method is complete and efficient, we achieved good performance with a reasonable computing budget.
>
> To test the scalability of our method to larger graphs, we conducted experiments on the FAUST dataset used in [1]. The data consists of 100 meshes of human bodies in various positions, and each mesh has 6890 vertices. Following the settings in [1], we report means and standard errors of the mean of over three runs. Results are:
>
> | **Method**| *Performance* |*Training time* |
> | ----------- | ----------- | ----------- |
> | SphereNet without 3D info|$77.42\pm2.7$%|12 mins|
> | SphereNet|$98.67\pm0.03$%|15 mins|
>
> Key points from results:
> 1. **Performace**. Even though SphereNet is not designed for meshes, it still achieves the performance of 98.67%, which is better than most baselines in [1], and a little worse than the proposed Gem-CNN.
>
> 2. **Speed**. Our computing infrastructure is 1 commonly used Nvidia GeForce RTX 2080 TI 11GB GPU. SphereNet only needs several seconds one epoch. The whole training process needs 140-150 epochs to converge and the total training time is around 15 mins. We do not find the training time of Gem-CNN and other baselines in the paper, but **the speed and efficiency of SphereNet on this data are totally acceptable**.
>
> 3. **3D Info**. If we don't use the 3D information, our SphereNet is simply downgraded to a regular GCN and the performance is much worse (77.42%). This implies the power of our proposed solution to consider complete geometrical information.
>
> **We can put the results on meshes in the Appendix of the paper if you suggest so.**
>
> We continue at the 2nd part as below:

---

### Official Review · Reviewer_xP3s · 2021-11-02

**Correctness:** 3
**Technical Novelty And Significance:** 4
**Empirical Novelty And Significance:** 4
**Recommendation:** 8
**Confidence:** 5

**Main Review:**


Strengths:

(1) The SMP introduces an interesting method to alleviate the computation cost issue in SCS from O(nk^3) to O(nk^2). This method is important and can be generalized to more broad types of tasks.

(2) The paper structure, i.e., from motivation to solution, is generally smooth. (Some potential points are listed below).

(3) This is an empirical work, and the experimental results support the effectiveness of SMP.


-----
Weaknesses:

(1) About motivation: Why SMP is better than Cartesian coordinate system (CCS) is not well explained. Authors give an example in Sec 2, but it is a little confusing for me.
1. It says atomic connectivity is based on a cut-off distance and `considering all pairwise distances will lead to severe redundancies`, thus only local distance pairs are considered. But my understanding is the cut-off distance is merely a hyper-parameter, and it can include the H-H bond in the example in Fig 1.
2. SMP alleviates the high computation cost issue when using SCS. However, its computational cost is still quite large, and is larger than the computational cost in CCS given the same cut-off threshold.
3. From my point of view, SMP can better reflect the chirality property where models using pairwise distance cannot. This might be one point authors can try.

(2) One question about the SMP model.
So SCS takes 3 atoms to define the reference plane. In SMP, it reduces it by introducing an interesting trick. However, I’m not quite sure how to pick the atom q_1 (in Figure b). Will the choice of q_1 have an effect on SMP? I didn’t find it in Sec 3.


(3) Categories in related work.
About the categories, I’m not sure why authors split the existing works into EGNN and relative information. From my understanding, models like SE(3)-Trans and EGNN are also using relative distance.



-----
Minor:

(1) Typo in Intro: `... relative location of each atom in a 3D graph … `

(2) In Intro, the authors say that `integrating the SMP and physical solutions to the Schrodinger equation`, can authors further explain what this means?


**Summary Of The Paper:**

This paper provides a novel geometric GNN: SMP. SMP is based on the SCS, and can reduce the complexity by an order of magnitude. Besides, it is a unified framework that can cover some mainstream geometric GNN methods. The empirical results support the effectiveness of SMP.

**Summary Of The Review:**

This paper is generally good, in terms of story-telling, method description, and technical novelty.
There are three main concerns: motivation/example, model description, and related work. This work can be further polished after fixing these points.

-----

After checking the replies from the authors and polished-up version of the paper, most concerns on the story have been addressed. I would like to raise the score to 8.

---

> ### Author Response · Authors · 2021-11-11
> **Updated the motivation why using SMP rather than CCS. Cut-off is not for CCS but for SCS, and using all pairwise distances is not for comparing SMP and CCS, but for determining what relative info is used in SMP. Questions are answered and related work is updated in the revision. (2nd part)**
>
>
>
> > Explain "integrating the SMP and physical solutions to the Schrodinger equation"
>
> We updated the intro. As stated in Sec. 4, Ψ(d,θ,φ) is one solution to the Schrodinger equation, which imposes physically meaningful representations (vectors) for d, θ, and φ. SMP provides the justification of approximate completeness from the geometry perspective. Integrating these two could lead to approximately complete and meaningful representations of 3D molecular graphs.
>
>
> >Reference
>
> [1] Tensor field networks: Rotation- and translation-equivariant neural networks for 3D point clouds
>
> [2] Equivariant message passing for the prediction of tensorial properties and molecular spectra
>
> [3] SE(3)-Transformers: 3D Roto-Translation Equivariant Attention Networks

---

> > ### Comment · Reviewer_xP3s · 2021-11-13
> > **Follow-up questions**
> >
> > First, thank the authors for the reply. I still have some confusing points, just trying to understand.
> >
> > > About SMP and CCS.
> >
> > Thanks for adding the discussion on EGNNs and SphereNet. Actually, the authors, to some extent, misunderstood my question. I was asking about the SMP and CCS, and the authors' replies are the difference between SMP and CCS in modeling. The sentences added in Sec seem enough to complete this motivation.
> >
> > But one thing to point out is, chirality is a good point, but adding it without empirical proof can make the story a little weak. I'll leave this question for the authors also.
> >
> > - BTW. I already tried SphereNet on the chirality task, but not working as expected (almost the same as SchNet, DimeNet++).
> >
> > > About q_1.
> >
> > The authors misunderstood my question. I get the point of Fig 2, which from my point of view, is the most interesting insight from this paper. Actually what I want to ask is, choosing different atoms for q_1, will this change the message passing results? The authors say no, so I'm wondering how? Is this because of using summation for the aggregation function? If so, authors can explicitly add it in Sec 3.
> >
> > > Schrodinger equation
> >
> > Thanks for answering this. Now I have another question. So Schrodinger equation is used to model the electrons, while SMP models the atoms, not electrons. Can authors help explain the discrepancies?
> >
> > I'm aware that some previous papers also have this claim, so this won't affect the scoring.
> >
> > --------
> >
> > Since now the story is more complete, plus after checking the reviews from other previous top-tier papers on 3D GNN, I will raise the score.

---

> > > ### Author Response · Authors · 2021-11-19
> > > **Responses to follow-up questions**
> > >
> > > We are happy to know that most previous concerns were addressed in the revision and responses. Here are responses to follow-up questions:
> > >
> > > > About SMP and CCS.
> > >
> > > We are happy to know that the statements added in Sec 2 can fully complete the motivation you asked. The purpose of the replies was to provide some background for a better understanding of the statements in Sec 2. But now we realize that the sentences in the papers are already enough, thanks!
> > >
> > > > Chirality theory and empirical proof.
> > >
> > > Thanks for bringing this to our attention.
> > > First, we think chirality is a good point in theory, as chirality is the closest relationship between two different molecules in nature - same atoms, same edges, same angles between the edges, and the only difference is the asymmetry that one is the mirror image of the other. Methodology-wisely, previous methods like SchNet or DimeNet can not distinguish but SphereNet can. We conducted experiments on commonly used datasets for molecular representation learning, including QM9, MD17, and OC20. Our assumption was: in those large datasets containing a lot of molecules, there might be two stereoisomers (chiral) existing and they must exhibit different properties, and our SphereNet is able to distinguish and learn it. But yes, we did not conduct experiments for chirality specifically.
> > >
> > > There is a paper currently under double-blind review that designs experiments specifically for chirality learning. They conduct two sets of experiments to show that SphereNet is more powerful than SchNet or DimeNet++.
> > >
> > > The first experiment: contrastive learning to distinguish stereoisomers. They use a subset of the PubChem3D dataset and it contains 2.98M conformers for 598K stereoisomers. They use two conformers from the same stereoisomer as a positive pair, and two conformers from different stereoisomers as a negative pair. They visualize the clustering results in a learned latent space to see if the model is capable of clustering conformers sharing the same chiral molecule. They show that SphereNet achieves almost the perfect separation between stereoisomers, and is much better than SchNet and DimeNet++.
> > >
> > > The second experiment: classifying chiral centers to R/S. They still use a subset of the PubChem3D dataset and it contains 466K conformers for 78K stereoisomers with one chiral center. The task is to classify chiral centers to R/S and the result is:
> > >
> > > | **Method**| *R / S Accuracy (%)* |
> > > | ----------- | ----------- |
> > > | SchNet|$54.5\pm0.2$%|
> > > | DimeNet++|$65.7\pm2.9$%|
> > > | SphereNet|$98.2\pm0.2$%|
> > >
> > > From their results we can see SphereNet is nearly perfect and much better than SchNet or DimeNet++.
> > >
> > > They also have a third experiment for chirality property prediction but only SphereNet is used. We cannot reveal the paper information, but **will add more discussions on this (chirality and empirical studies) later on in the final version** after the double-blind review process. We also wish to work on the chirality task from your side in the future.
> > >
> > > >About q_1.
> > >
> > > Sorry for the misunderstanding, now it's clear. Yes, you are correct. It will not change the message passing results, because we do summation when aggregating information from its neighborhood for an atom. **We have revised the paper and explicitly added it in Sec 3**.
> > >
> > > >Schrodinger equation.
> > >
> > > Thanks for initializing this interesting discussion on the Schrodinger equation and its application in molecular learning. We are not experts but have been thinking about this intensively and talked to some experts recently. Here are our best understandings so far:
> > >
> > > First of all, in quantum physics, electron density is used to describe 'position' of an electron and it measures the probability of an electron being present at an infinitesimal element of space. Then an electron density with the electron wave function is the key and its solution space is given by the Schrödinger equation.
> > >
> > > But originally, the separation of variables in polar coordinates (d, α, ϕ) is for describing geometry between one electron and the atom. For atom-level molecular ML studies, they don't have a well-defined solution to study atom-atom geometry. Hence, in solution space of Schrödinger equation, they just replace the electron by another atom. By doing this, an inductive bias from quantum mechanics is added to formulate the relative position between two atoms, somehow resulting in meaningful atom-level representations. For sure, we don't think this is the most perfect solution, but the feasible one up to now. Currently, we are also working to explore this more deeply. We read Chapter 4 of the book [1] when understanding this, and we are happy to continue the discussion in the future!
> > >
> > > >Reference
> > >
> > > [1] Griffiths, D. J. and Schroeter, D. F. Introduction to quantum mechanics, 3 edition. Cambridge University Press, 2018.

---

> > > > ### Comment · Reviewer_xP3s · 2021-11-20
> > > > **Follow-ups**
> > > >
> > > > Cool, thank the authors for mentioning that paper, and also sharing the book. I'll keep an eye on this.

---

> ### Author Response · Authors · 2021-11-11
> **Updated the motivation why using SMP rather than CCS. Cut-off is not for CCS but for SCS, and using all pairwise distances is not for comparing SMP and CCS, but for determining what relative info is used in SMP. Questions are answered and related work is updated in the revision. (1st part)**
>
> Thank you so much for your detailed and constructive comments! We have revised the paper substantially and also provide responses here.
>
> > Motivation: why use SMP rather than CCS
>
> We think there might be some misunderstandings.  Actually, EGNNs usually take absolute 3D info defined in the Cartesian coordinate system (CCS), and our method takes relative info like distance and angle. This is the difference between EGNNs and our method.
>
> **To explain the motivation of SMP rather than EGNNs in CCS, let’s start with how EGNNs work with absolute info.** We assume a input 3D molecule contains two atoms: A (0,0,0), B(1,1,1). EGNNs directly take such absolute 3D info (xyz coordinates). Imagine we rotate this molecule in CCS to new xyz coordinates: A(0,0,0), B(-1,-1,-1).  Literally, the coordinates change, the learned molecule representation changes, then the molecular properties change. But this is not true! Rotation should not change the molecule and its properties. Now, this is the key of EGNNs: we need to carefully design network components to be equivariant. Thus, rotating inputs would lead to the equivariant output. The key idea of EGNNs like TFN [1] and SE(3)-Transformer [2] is to put some constraints on kernels and nonlinear functions to make them equivariant. This essentially implies the key difference between EGNNs (in CCS) and SMP. Our SMP represents another category that takes relative 3D info such as distance. In the above example, the distance of AB is sqrt(3) and rotation would not change it. Hence, it’s naturally invariant. We can use regular message passing methods to incorporate such relative info for rotation invariant outputs. Our work is to give answers on: What relative info should we use? How to incorporate such info? For what purposes?
>
> **Now let's discuss the motivation of using SMP instead of CCS, which we update in Sec 2 & Sec 5.1 and also summarize as below.** Firstly, EGNNs need the careful design of every network component to be equivariant, which is expensive. For example, for TFNs, the computational budget includes the computing of spherical harmonics and Clebsh-Gordan coefficients, a ‘for’ loop on different output types for one specific input type, etc. Secondly, for molecular learning, the system is usually required to be invariant, hence, the complicated SE(3) group representations in EGNNs are not necessary. Thirdly, the performance of EGNNs is worse than SMP, which is shown in our experiments.
>
> > Motivation: why SMP, not all distances
>
> **A misunderstanding here is: whether using all distances or not is NOT for comparing SMP and CCS, but how to wisely incorporate relative info in SCS. We update the 2nd paragraph of Sec 2 for clarification.** Yes, we use both real chemical bonds and cut-off distances as the distance info, and cut-off distance is a hyperparameter. Using a small cutoff may not make H-H an edge in Fig. 1. Also, in a large molecule, most pairwise connections may not be modeled as edges. Even though we assume we can set a large cut-off threshold, making all pairwise connections as real edges. the consequences could include: the molecular graph is too complex, making the training suffer; the model gets confused about which info is useful and which is not, leading to bad performance. Also, as you said, **it can not recognize chirality.** Hence, we conduct rigorous analyses, then design SMP to incorporate angle and torsion for approximately complete molecule graph representations. By doing this, we can achieve the benefits of theoretical completeness, low complexity, easy training, and superior performance. The chirality property is just one example that SMP can distinguish but using all pairwise distances cannot. There must be any others, for SMP can theoretically guarantee the approximate completeness while the uncovered cases may not exist.
>
> > How to pick the atom q_1 in Fig 2(b)
>
> Any atom could be q_1, which will not influence the result. As described in Sec. 3.1, the corresponding torsion angle of each atom only depends on its previous atom in the anticlockwise direction. Hence, any atom could be q_1. q_1, q_2, and q_3 in Fig are just for a clear illustration.
>
> > Categories in related work and SE(3)-Trans also use relative distance
>
> You are true that EGNNs use relative info! thanks for the valuable reminder! We have revised the category and several statements in Sec. 5.1. The only noteworthy thing is: the raw input of EGNNs is usually a vector **v** in CCS, and it's decomposed into orientation **v**/|**v**|, and length |**v**| later on. This is because spherical harmonics used in EGNNs are defined in S^2, which can only recognize orientation. Thus, the relative info like length needs to be considered separately for complete representations [1,2]. A similar situation also exists in Eq.(8) in [3].
>
> **Other responses and reference are included in the 2nd part as below.**

---

### Official Review · Reviewer_epnb · 2021-11-03

**Correctness:** 3
**Technical Novelty And Significance:** 3
**Empirical Novelty And Significance:** 2
**Recommendation:** 5
**Confidence:** 5

**Main Review:**

Strengths:
- The idea of incorporating torsion information when representing 3D molecules is novel and helpful. The proposed MP approach can better distinguish certain structures than some existing models.
- The proposed model performs well on benchmark datasets, which validate the strong expressive power when representing 3D molecules.

Weaknesses:
- The proposed SMP scheme in Eq. (1) lacks novelty since it basically enriches the GN framework in [1] with geometry features.
- I feel that the architecture of the proposed SphereNet is similar to DimeNet [2] except addressing torsion information. The importance and power to embed both distances and angles in 3D molecules have already been addressed by DimeNet. The ways of representing and embedding distances and angles are nearly the same in SphereNet and DimeNet. By comparing Figure 4 in this work and Figure 4 in DimeNet work [2], it's not hard to find the corresponding similarities in their architectures (each row below represents a correspondence):
+---------------+----------------+

|    DimeNet  | SphereNet    |

+---------------+----------------+

| Embedding | Input Block        |

| Interaction | Interaction Block |

| Residual    | Residual              |

| Output      | Output Block         |

+---------------+----------------+


[1] Relational inductive biases, deep learning, and graph networks

[2] Directional message passing for molecular graphs

**Summary Of The Paper:**

This work aims at the representation learning of 3D molecular graphs and proposes a principled message passing framework to unify existing works. Besides, a novel message passing approach is presented by incorporating 3D geometric information including distance, angle, and torsion. In experiments, the performance of the proposed model is validated on three benchmark datasets.

**Summary Of The Review:**

The main issue of the proposed model is the lack of novelty. Although introducing torsion is helpful when representing 3D molecules, the proposed model doesn't have novel architecture when embedding it as compared to existing models. The proposed framework also just follows the idea of existing work. Thus I think this work is below the acceptance threshold of ICLR.

---

> ### Author Response · Authors · 2021-11-10
> **GN framework is just where we start from, we focus on P in Eq.(1)  and removing other notations would not affect main claims, main novelty, even the main content of our paper; our novelty is on new message passing scheme and learning 3D info completely and efficiently; papers in this research line use similar architectures, and DimeNet also follows PhysNet.**
>
> Thanks a lot for your constructive comments! We have revised the manuscript accordingly and also provide responses here.
>
> > SMP scheme in Eq. (1) and the GN framework in [1] appear as similar
>
> **In short, the GN defines 2D graphs and is naturally a starting point. Discussing 3D info *P* is the main content in this research line and our paper.**
>
> Regarding the framework in Eq. (1), we merit the existing work [1] for 2D graphs and make it our starting point. But the key difference is the 3D info *P*, **which is our focus in the entire paper**, and also the focus in this research line like [2,3,4,5]. Basically, Our work shows how to define *P*, how the existing studies (SchNet [3], DimeNet [2]) deal with *P*, how to involve *P* with a new message passing method (SMP), how to incorporate *P* completely and efficient, etc. **Hence, in short, we are discussing *P* and 3D info in this work. Removing other parts and notations in Eq. (1) will not affect the main novelty, main claims, even the length and content (a little bit) of this paper.** The reasons why we adopt the GN in [1] are: it's for 2D graphs and can naturally be where we start from; we think it’s an elegant framework, and we want to merit it, start from it, and unify our work to this research line. **We absolutely value your concerns and add several statements at the end of Sec. 3.1 to make this clear**.
>
> >Main novelty of this work – how to incorporate 3D info *P* completely and efficiently
>
> We focus on the important 3D info *P* in this work and formally define the 3D graphs. Our main novelty sketches around this point. We emphasize our novelty here. Importantly, we want to point out **the motivation of SMP is not simply including extra geometry (like torsion), but the complete and efficient representations for 3D molecular graphs without information loss.** To achieve this completeness, the original way of computing torsion induces high complexity, as analyzed in Sec 3.2. Thus, we design a new solution to compute torsion, tremendously reducing the complexity and enabling the application of SMP to large molecules (like catalyst materials). Our method is approximately complete and the uncovered cases may not exist in nature. In short, we believe our novelty is important and significant, as **methodologically, it studies completeness of 3D graphs without information loss, and is implemented in an efficient and scalable way; practically, it's the best model so far for various tasks across small and large molecules**.
>
> >The used network architecture is similar to DimeNet [2]
>
> We agree that we use similar network architecture as DimeNet [2]. Actually, starting from SchNet[3], important papers in this research line always use this pipeline, which roughly consists of embedding blocks, interaction blocks, and output blocks, etc. **For example,
> DimeNet (Fig. 4) follows PhysNet [4] (Fig. 1) and they use very similar architecture, which is consist of embedding, interaction, residual, and output blocks**. Other studies like SchNet [1], GemNet [5], and PaiNN [6] also use similar architectures. Each of the methods makes distinguishing contribution to the community own to its novel message passing method for incorporating 3D info. **Hence, we use similar architecture for consistency and fair comparisons**.  We argue the main contribution of our SMP is a new message passing solution to study completeness with considering efficiency. We add a statement in the last paragraph of Sec 4 to make it clear to readers.
>
>
> >Reference
>
> [1] Relational inductive biases, deep learning, and graph networks
>
> [2] Directional message passing for molecular graphs
>
> [3] SchNet: A continuous-filter convolutional neural network for modeling quantum interactions
>
> [4] PhysNet: A Neural Network for Predicting Energies, Forces, Dipole Moments, and Partial Charges
>
> [5] GemNet: Universal Directional Graph Neural Networks for Molecules

---

> > ### Comment · Reviewer_epnb · 2021-11-19
> > **Still some concerns about novelty**
> >
> > Thank you for the authors' feedback! I have read it and the other reviews. The authors claim that the novelty is on new message passing scheme and learning 3D info completely and efficiently. However, I still have some concerns about it:
> >
> > - About the completeness
> >
> > For this part, I agree that the proposed method based on SCS can (approximately) represent complete 3D info, which contributes to the novelty. While I'm not surprised since there are existing works that try to represent 3D info (especially 3D point clouds) based on SCS [1] [2]. From the same motivation, the proposed message passing scheme (SMP) can be seen as a different way to achieve it. However, as I will discuss in the section below, the technical contribution of SMP is limited.
> >
> > - About the similarity between SphereNet and DimeNet
> >
> > Although the related works including SchNet, PhysNet, DimeNet, etc, have similar pipelines, I'm focusing on the detailed computations and formulas inside the blocks instead of just the names. With similar pipelines, the previous works all have significant technical contributions when designing the blocks.
> >
> > For example, I would like to compare the message passing schemes in PhysNet and DimeNet: PhysNet doesn't take angles in its message passing scheme in the Interaction block, and the message passing is operated on only one-hop neighbors around nodes in each iteration. While DimeNet proposes a novel and workable way to define and encode the angular info in message passing. To do so, the message passing in DimeNet is operated on two-hop neighbors to capture angles, and there are both summations of messages in the Interaction block and the Output block to implement the unique two-hop message passing. DimeNet also proposes novel basis functions for encoding distances and angles.
> >
> > However, when comparing the message passing schemes in DimeNet and SphereNet, I find that SphereNet still follows the same two-hop message passing scheme as in DimeNet. SphereNet just further uses an element-wise multiplication and two linear layers to encode the torsion angles in the messages, which is already the same approach in DimeNet to encode angles. Although the message passing scheme in SphereNet can completely represent 3D info, it only uses the existing workable scheme and way to encode angles. Thus I feel that the technical contribution is limited.
> >
> > - About the efficiency
> >
> > I think SphereNet is not efficient enough and does not have enough potential to be applied to macromolecules. As shown in Table 9 in the paper, SphereNet requires more # of parameters and time cost per epoch than DimeNet++, which is a model with O(N^2) complexity (where N is the number of neighbors as analyzed in PaiNN [3]). In [3], the scaling issue of the models (e.g. DimeNet/DimeNet++) explicitly using angles to represent 3D info has been clearly demonstrated. In my own experiments before, DimeNet/DimeNet++ indeed have the issue when applied to macromolecules (e.g. with hundreds of atoms). With restricted resources, the cutoff distance cannot be set to a large enough value (e.g. 10 angstrom) to avoid the out-of-memory issue. However, a large cutoff is usually needed to capture the long-range interactions. Since SphereNet uses a very similar message passing scheme as DimeNet/DimeNet++ and also explicitly encodes angles, the scaling issue is unsolved and cannot be convinced to be novel enough regarding efficiency.
> >
> > As a comparison, PaiNN proposes a much more efficient way to represent 3D info with only O(N) complexity. The # of parameters required by PaiNN is only 1/3 of DimeNet++ and the inference time of PaiNN is only 29% of DimeNet++. So that PaiNN is significantly more efficient than SphereNet. While being efficient, PaiNN can outperform SphereNet on 6 targets on QM9. Overall, I'm not impressed by the proposed model.
> >
> > [1] Spherical kernel for efficient graph convolution on 3d point clouds
> >
> > [2] Modeling Local Geometric Structure of 3D Point Clouds Using Geo-CNN
> >
> > [3] Equivariant message passing for the prediction of tensorial properties and molecular spectra

---

> > > ### Author Response · Authors · 2021-11-20
> > > **Responses to Reviewer epnb on concerns about novelty (3rd part)**
> > >
> > > > About the efficiency.
> > >
> > > Thanks for emphasizing PaiNN, which is an excellent work in the area. We have been paid intensive attention to PaiNN and it is one of Equivariant GNNs and was included in our related work Sec 5.1.1. We also included the motivations of using invariant GNNs instead of EGNNs at the beginning of Sec 2. PaiNN takes a 3-dim vector (atom position) as input and similar methods include ForceNet [1] and NequIP [2]. This is why complexity of PaiNN is O(N). **In short, PaiNN represents one direction for molecular learning-Equivaruiant GNNs (Sec 5.1.1), and SphereNet represents the other-Invariant GNNs (Sec 5.1.2). PaiNN is an approximation (only uses type-1 basis) of full spherical harmonics, which is the reason why it's fast but the performance is not the best. On QM9, the overall std MAE of SphereNet (0.91) is much better than that of PaiNN (1.01).**
> > >
> > > ***Performance***: As we stated in Sec 5.1.1 and Sec 2, SphereNet achieves superior performance compared with EGNNs. On QM9, even though PaiNN has 6 best results, SphereNet achieves 5 best as well as 3 second best. **More importantly, for the overall performance on all the 12 properties, SphereNet has the best std MAE of 0.91, which is much better than PaiNN (1.01). PaiNN is also worse than DimeNet++ (0.98).** Among all properties, HOMO-LOMO gap is the most important one and SphereNet is 31.1, which is much better than PaiNN of 45.7. On the large-scale dataset like OC20, PaiNN has no results, but the similar work ForceNet is worse than DimeNet ++, and DimeNet ++ can not compare with SphereNet.
> > >
> > > ***Methodology and efficiency***: PaiNN (and ForceNet, NequIP) represents one research direction on molecular learning and it is Equivariant GNNs (Sec 5.1.1). SphereNet (SchNet, PhysNet, DimeNet) represents another one that is invariant GNNs. The former takes absolute 3D information (original xyz coordinates, you can also see details about how EGNNs work with the xyz coordinates in our responses to reviewer xP3s), thus needing carefully designed equivariant layers to make the learning valid. This actually puts hard constraints on each network component. For example, the conv kernel needs to be the result of learnable parameters product spherical harmonics; the common non-linearities like ReLU cannot be used, and non-linearities also need careful design. One point is for molecular learning, the downstream tasks usually need the representation to be invariant, and the complicated equivariance may not be necessary. In addition, the reason why PaiNN is fast is - it only uses the type-1 basis in Spherical harmonics, which is an approximation. It's proved in TFNs [3] that theoretically, $\ell$  should be infinite, and in practice, the type $\ell$ should be up to 2 for achieving satisfactory performance. Type-1 basis essentially constraints the conv kernel to be in a narrow learning space. This is why PaiNN is fast yet the performance is not the best. However, when using type-2, the conv kernel can be more expressive but the efficiency could be a bottleneck. That's why we study another direction:invariant GNNs.
> > >
> > > We value PaiNN and other studies(ForceNet, Nequip) representing an interesting research field in molecular ML. SphereNet along with some works represents another direction, and it's invariant, complete, straightforward, and has the best performance. We admit efficiency is an issue, and complexity for SphereNet and DimeNet is O(N^2), as you said. At least, SphereNet is complete but not slower than the previous methods. It could serve as a foundation in this research line of invariant GNNs, based on which it's also promising to develop an efficient method that is fast (o(N)), complete, and has the best performance.
> > >
> > > We sincerely thank you for your time! We are really happy to make everything super clear through such discussions. We look forward to your reply and further discussions, thanks!
> > >
> > > > Refrence
> > >
> > > [1] ForceNet: A Graph Neural Network for Large-Scale Quantum Calculations
> > >
> > > [2] SE (3)-equivariant graph neural networks for data-efficient and accurate interatomic potentials
> > >
> > > [3] Tensor field networks: Rotation- and translation-equivariant neural networks for 3D point clouds

---

> > > ### Author Response · Authors · 2021-11-20
> > > **Responses to Reviewer epnb on concerns about novelty (2nd part)**
> > >
> > > > About the similarity between SphereNet and DimeNet in architecture (architecture of SphereNet is in Appendix).
> > >
> > > In summary, we want to emphasize in the first place that we propose to **add the torsion for complete learning, and the regular torsion needs 3-hop information. We narrow it into 2-hop, which is one of our main novelty: complete and efficient**. In addition, **Performing within *n*-hop neighborhood or doing *n* sum operations (*n*=1,2) is not decided by a particular network, but by what information is used**. For SchNet and PhysNet, *n*=1, because they only use distance info. DimeNet and GemNet-T[1] use angle info such that *n*=2.
> > >
> > > Performing within *n*-hop neighborhood or conducting *n* sum operations is not due to it's PhysNet or DimeNet, but decided by what 3D information is used. SchNet and PhysNet both use edge info, hence, they naturally consider 1-hop information. As the final objective is to achieve the node feature, hence, they both use only 1 sum operation for aggregation (SchNet Fig. 2 interaction block, PhysNet Fig. 1 interaction block).  DimeNet and GemNet-T both incorporate angle information, hence, they naturally consider 2-hop neighborhood. Similarly, the final objective is to achieve the node feature, hence, they both use 1 sum for aggregating information to edges (messages) then another sum to aggregate information from edges to the node (DimeNet Fig. 4 interaction block and output block, GemNet-T Fig. 2 interaction block and a final sum without explicitly named as output block but it's similar). Hence, how to perform n-hop information and how to do sum are natural. SchNet and PhysNet are similar as they consider edge, and DimeNet and GemNet-T are similar as they consider angle.
> > >
> > > The original implementation of torsion is 3-hop. If following the pipeline from SchNet(PhysNet)--->DimeNet(GemNet-T), SphereNet may use 3-hop information and perform 3 sum operations for achieving node features! But we design a novel way to compute torsion with only 2-hop information, making it much more efficient as well as approximately complete.
> > >
> > > In the network architecture, we indeed use linear blocks for incorporating edge, angle, and torsion as same as DimeNet++. We also use the same flow that performs element-wise multiplication from distance representation, angle representation, to torsion representation sequentially, because this flow is straightforward and elegant. For basis, we tried a lot of basis functions including MLP, Gaussian, Sine, spherical Bessel, spherical harmonics, Berstain, etc when running experiments. We found spherical Bessel is the best for encoding distance, and spherical harmonics is the best for encoding angle and torsion. Thus, we used them in our network design. But we still want to emphasize that the main difference from DimeNet is a new MP scheme with a novel torsion for complete and efficient learning, which is also the main content of our paper.
> > >
> > > >Reference
> > >
> > > [1] GemNet: Universal Directional Graph Neural Networks for Molecules
> > >
> > > We continue at the 3rd part as below:

---

> > > ### Author Response · Authors · 2021-11-20
> > > **Responses to Reviewer epnb on concerns about novelty (1st part)**
> > >
> > > Thanks a lot for your valuable comments! We really appreciate it and provide responses here:
> > >
> > > > About motivation
> > >
> > > **We are happy to know that you agree with one of our main novelty: completeness**, which is achieved by a new solution to compute torsion. We also sincerely thank you for bringing work on 3D point clouds to our attention, but they are totally different from SphereNet on motivation, novelty, technical solutions, etc.
> > >
> > > ***Motivation***: **methods on 3D point clouds are translation invariant, while SphereNet on molecules is rotation invariant**.
> > > We want to emphasize that SphereNet is driven by totally different motivations. The molecular learning needs the input to be invariant (one research line as in related work Sec 5.1.2) or networks to be equivariant (the other research line as in related work Sec 5.1.1). Both the work [1] and [2] on 3D point clouds design networks to be translation invariant. We believe in ML, the problem itself is the most important thing and the method should be appropriately designed for solving the problem. In SphereNet, we **build a local SCS for each node**, then use the relative information (edge, angle. torsion) such that our method is naturally invariant, which is important for downstream tasks like property prediction on molecular learning. However, the mentioned two studies use **ONE global coordinate system (either CCS or SCS) for ALL points**, through which rotation invariance is not feasible. They instead put efforts to achieve translation invariance on 3D point clouds, which can be valid for objective detection and semantic segmentation, etc.
> > >
> > > ***Motivation and Novelty-completeness***. **Studies [1] and [2] are not complete due to the discretization in spatial space**. This is a big difference with the research line in molecular learning that the learning space is continuous. For example, SchNet uses a continuous-filter convolution to encode distance. **The consequences of discretization could be twofold. (1). Not complete**. the space is split into several grids, and two nodes in the same grid will exhibit the same geometrical representations. Hence, this is not complete, especially if the grid is large that two distant nodes would be modeled as the same, leading to severe information loss. Specifically, work [1] splits a sphere (defined by a cutoff for each node) into several grids by performing discretization along (r, α, ϕ) directions. Work [2] does discretization along r direction by combining different radius cutoff, along with three angles in x, y, z directions. Both methods can not specify the position of each node, thus, some 3D information is lost. Our SphereNet is able to determine the position of each atom in 3D space, making it more expressive. **(2). Efficiency issues**. Discretization in spatial space may lead to information redundancy, as a lot of grids may not have valid nodes inside, then the data is sparse, making the training hard.
> > >
> > > **Beside the important invariance and completeness, there also exist** ***several main technical differences***. The mentioned two methods use node aggregation to update node features from neighboring nodes, it's more like a graph conv operation. We follow the message passing scheme for molecular learning that performs edge aggregation to update edge features (messages) from neighboring edges. In addition, Since the networks are originally translationally invariant,  they conduct data augmentation to include multiview information of point clouds, somehow achieving 'multiview invariant'. This is a commonly used technique in ML that uses data augmentation to approximate invariance, but data augmentation is expensive and can not guarantee learning invariance successfully.
> > >
> > > Although some works on 3D point clouds are totally different from SphereNet (also other studies on molecules like SchNet and DimeNet) from perspectives like motivation, novelty, and techniques, we are really happy to make everything super clear through thus discussions. We also thank the reviewer for bringing the work on point clouds to our attention, as we now think discretization in spatial space could be a good idea to be applied to some hard research problems in molecular learning.
> > >
> > > > Reference
> > >
> > > [1] Spherical kernel for efficient graph convolution on 3d point clouds
> > >
> > > [2] Modeling Local Geometric Structure of 3D Point Clouds Using Geo-CNN
> > >
> > > We continue at the 2nd part as below:

---

> > > ### Author Response · Authors · 2021-11-23
> > > **Look forward to your reply and further discussions**
> > >
> > > Dear Review epnb,
> > >
> > > First, we are happy to know you agree that completeness (with a new torsion) contributes to our novelty. Regarding your remaining concerns on novelty, we made great efforts and provided detailed responses with in-depth analyses and comprehensive discussions.
> > >
> > > For each of your concerns, we not only referred to the appropriate place of our paper for clarification, but also used space adding extra introductions on the background with the purpose of clearer illustration and better understanding. For each of the papers you mentioned, we not only clarified how each specific work differs from our method on motivation/novelty/technical solutions, but also reviewed and summarized the whole research category each paper represents and compared it with our method from different views. Similar literature review is also provided in the related work of our paper. Some studies are excellent even though they are from different domains (like 3D point clouds). Their limitations motivated our method, and their strengths can inspire great ideas to solve some other research problems in molecular learning.
> > >
> > > We hope we addressed all of your concerns in those responses. As mentioned in the response, we are also excited to make everything super clear through such discussions. We look forward to your reply to see if you still have remaining concerns, and we are also happy to continue such fruitful discussions.
> > >
> > > Thanks for your time!

---

> > > > ### Comment · Reviewer_epnb · 2021-11-24
> > > > **Thank you for your response**
> > > >
> > > > First, I’d like to thank the authors for the detailed feedback addressing my concerns. I’m impressed by the motivation and solution to (approximately) achieve completeness of 3D graphs as the authors have more clarifications about works on 3D point clouds. The way of computing 3D information including distance, angle, and torsion from the 3D graphs is novel and helpful.
> > > > Regarding the exact approaches including the proposed message passing scheme and the GNN that implement the solution to achieve completeness, I still feel that their ideas are guided and constrained by existing methods. Although the used 3D information would affect the design of message passing schemes over n-hop neighborhood, I would expect novel message passing schemes to incorporate 3D information. As replied by the authors, they think it’s natural to use the existing way of considering 1-hop information when considering edge (or pairwise distance), and it’s natural to use the existing way of considering 2-hop information when considering angle. So that they still follow those ideas to incorporate torsion information. Although it is totally workable to perform message passing scheme in this way, the resulting approach is still in the fashion of existing works and has the same issue regarding the O(N^2) complexity. Currently in this field, whether there exist other more powerful and efficient ways of using 3D information in invariant GNNs for 3D graphs is unknown and needs to be investigated. Thus I would be more impressed by the work if it brings any novel ways or demonstrates that it is only possible to use existing ways (e.g. using 2-hop information when considering angle).
> > > >
> > > > Regarding the efficiency, the proposed SphereNet is not slower than the previous invariant GNNs as stated by the authors. However, it still suffers from O(N^2) complexity and is not more efficient than the Equivariant GNN, PaiNN, which has competitive performance in related tasks. Thus I feel that the claimed efficiency is only meaningful in a relatively limited scope.
> > > >
> > > > Overall, I acknowledge the work has novelty regarding the motivation to achieve completeness of 3D graphs and the solution of computing 3D information. However, I think the work needs to be improved to meet the acceptance threshold of ICLR by discovering a novel way to incorporate the 3D information beyond the existing fashion or demonstrating that the existing fashion is the only possible approach.

---

> > > > > ### Author Response · Authors · 2021-11-25
> > > > > **Thanks for the reply; motivation makes sense; novelty is clear; the main additional concern is explained.**
> > > > >
> > > > > Dear Reviewer epnb,
> > > > >
> > > > > Happy holiday!
> > > > > We are excited to know now you think motivated by completeness, our solution of computing 3D information including distance, angle, and torsion from the 3D graphs is novel and helpful.
> > > > >
> > > > > > Additional main question: current flow is the only fashion?
> > > > >
> > > > > In your new response, now the additional question seems that you worry about the network is constrained by existing flow. Your example is if using 2-hop information is the only way to encode angle.
> > > > >
> > > > > 1. **2-hop for edge is a must**
> > > > >
> > > > > Our short answer for the example question is **yes! using 2-hop information is the only way to encode angle**. Even though we think SphereNet may inspire faster versions of invariant GNNs, to our best knowledge, an angle involves two edges. Then for the center node, its 2-hop information is necessary. This point is demonstrated in Invariant GNNs like DimeNet. Even in one Equivariant GNNs SE(3), when using attention to implicitly consider angle in the kernel, it also considers 2-hop information.
> > > > >
> > > > > 2. **Using all the three geometries is based on completeness analysis**
> > > > >
> > > > > For message passing methods to incorporate 3D info, we have conducted a comprehensive literature review and concluded there exist two categories: Invariant and Equivariant GNNs. For invariant GNNs, relative 3D information (distance, angle, torsion) is a must. Then we want to answer this question in our paper: what kind of relative information is needed and how to incorporate it for complete representation? Hence, as in Sec 2, we propose to conduct analyses in the SCS and find all the distance, angle, and torsion are necessary for completeness. So we are not actually constrained by the current fashion. SchNet, DimeNet, PhysNet, or GemNet-T does NOT analyze completeness in SCS. **Assume a case that if we found angle is not a must for completeness, we would not use it. Hence, our network design is totally driven by the rigorous analysis in Sec 2 and Fig 1**.  When we said "if we follow the idea of edge and angle for torsion", we were talking about how to design torsion, as regular computing needs 3-hop info. Our use of the three types of geometries is based on rigorous analysis in Sec 2 for completeness.
> > > > >
> > > > >
> > > > > 3. **Order of including distance, angle, and torsion in the message passing flow does not matter**
> > > > >
> > > > > The order of including distance, angle, and torsion in the flow does not matter. Because for each neighboring node, its final embedding is obtained by element-wise multiplication of its own distance, angle, and torsion embeddings, and order does not change the result. We just use a natural way involving distance, angle, and torsion sequentially.
> > > > >
> > > > > 4. **An efficient way for torsion**
> > > > >
> > > > > In terms of technical solutions to encode angle and torsion, as stated above, both common knowledge and literature make it clear that 2-hop is NECESSARY for angle. More than 2-hop (like 3) can be used for encoding torsion, but we find 2-hop is enough and efficient. GemNet-Q uses 3-hop, and for large structures like proteins, 3 torsion angles are used for identification [1].  **Hence, we are confident that our method may not be the only fashion, but the most efficient invariant GNNs for completeness. This totally due to the use of torsion and how to implement torsion (2-hop for angle is fixed)**. In addition, As proposed in [2], message passing is a general framework, and a lot of notable graph methods can fit in including Graph Conv, Gated GNNs, GATs, Interaction Networks, DTNNs, even the Laplacian Based Method, etc. **We derive a new MP framework and it can describe specific methods that are promising on 3D GNNs, which will not change the truth that angle needs 2-hop, using torsion for completeness is novel, and torsion with 2-hop is efficient**.
> > > > >
> > > > > > Another issue on comparing with PaiNN
> > > > >
> > > > > Regarding another minor issue on efficiency compared with PaiNN, as we responded with details in previous responses, PaiNN is an EGNN. It is an approximation of Spherical harmonics and is a simple version of TFNs. It's fast but has constraints for kernel space, thus the performance on small datasets is not the best, and there are no results on large datasets.
> > > > >
> > > > > > Summary
> > > > >
> > > > > In summary, the inclusion of all the distance, angle, and torsion is due to the completeness analysis. The order of encoding them in the message passing flow does not change results. Our fashion as the most efficient one is totally due to the use of new torsion.
> > > > >
> > > > > We also wonder if Reviewers xP3s and 6Y5X could share their thoughts on the additional question raised by Reviewer epnb.
> > > > >
> > > > > > Reference
> > > > >
> > > > > [1] https://fabianfuchsml.github.io/alphafold2/
> > > > >
> > > > > [2] Neural Message Passing for Quantum Chemistry

---

> > > > > > ### Comment · Reviewer_epnb · 2021-11-29
> > > > > > **Thanks the authors for the feedback**
> > > > > >
> > > > > > Overall, I acknowledge the novelty related to the completeness, including the way to compute torsion and the analysis. I'm still not satisfied with the implementation of the completeness in the proposed GNN. Since the proposed way of encoding the 3D information is based on the existing method like DimeNet, and there are already a number of works (e.g. HMGNN [1], SIGN [2], GemNet-T [3]) using a similar way to encode angle-related information, I'm expecting to see novel approaches to bring 3D information (distance, angle, and torsion) in GNNs. Although using 2-hop (or actually 2 edges) is needed for an angle, there shouldn't be only one way to design the message passing scheme and encode 3D information.
> > > > > >
> > > > > > Due to the aforementioned reason, I'm personally worried about the technical contribution which is the design of the GNN. I'd like to change my option from marginally below the acceptance threshold to be fine with either accept or reject. However, since there is no score for this, I'd be okay with the options of the other reviewers and the chairs about the novelty point.
> > > > > >
> > > > > > [1] Heterogeneous Molecular Graph Neural Networks for Predicting Molecule Properties, ICDM 2020
> > > > > >
> > > > > > [2] Structure-aware Interactive Graph Neural Networks for the Prediction of Protein-Ligand Binding Affinity, KDD 2021
> > > > > >
> > > > > > [3] GemNet: Universal Directional Graph Neural Networks for Molecules, NeurIPS 2021

---

> > > > > > > ### Author Response · Authors · 2021-11-29
> > > > > > > **Thanks for the quick reply**
> > > > > > >
> > > > > > > Dear review epnb,
> > > > > > >
> > > > > > > Thank you so much for your reply!
> > > > > > >
> > > > > > > Regarding your concern on the architecture flow for implementing SMP, we would like to break the architecture into main components to see what else we can do to implement it in a different way.
> > > > > > >
> > > > > > > based on previous discussions, we can confirm
> > > > > > >
> > > > > > > 1. All the distance, angle, and torsion are needed (completeness analysis).
> > > > > > >
> > > > > > > 2. The order of including distance,  angle, and torsion does not matter.
> > > > > > >
> > > > > > > 3. The general message passing process is common: for the center message, information is aggregated and updated from all messages pointing to it. For each incoming message, its distance, angle with the center message, torsion with other incoming messages seem fixed. We do not see much flexibility in this.
> > > > > > >
> > > > > > > Then the remaining components we can think of could be:
> > > > > > >
> > > > > > > 1.  Linear blocks for encoding the three geometries. When conducting experiments, we tried both: the one with bottleneck and the one without bottleneck. We found the one without bottleneck (actually used in DimeNet++) performs faster and better. For sure this is a common setting in deep network design that in a linear block with two layers, the first layer performs down-projection and the second performs up-projection.
> > > > > > >
> > > > > > > 2. Element-wise multiplication for distance, angle, and torsion embeddings in interaction block. During experiments, we also tried element-wise addition, and found it is much worse than element-wise multiplication. We think this may due to the idea of "gate" that information flow is controlled during the supervised learning process.
> > > > > > >
> > > > > > > We agree that there might be other ways to encode 3D information, but the computational bottleneck is decided by torsion and we already implement the whole pipeline in the most efficient way. Besides torsion, we do not see much flexibility for the flow design. As we choose settings that are either common or generating the best performance, and we don't see much flexibility for other components, could you please consider removing the score penalty for this?
> > > > > > >
> > > > > > > Also, we talk a lot about GemNet-T in our paper. It uses a very similar flow with DimeNet and the key novelty is on the geometric message passing and its universality. Regarding the flow, it would be very good if the reviewer can provide us with some general directions to go, and suggestions for other architecture component designs are also warmly welcomed! Thank you for your time!
> > > > > > >
> > > > > > > Best,
> > > > > > >
> > > > > > > Authors

---

> > > > > > > > ### Comment · Reviewer_epnb · 2021-11-29
> > > > > > > > **Regarding an efficient way to encode 3D information**
> > > > > > > >
> > > > > > > > Indeed the current computational bottleneck is decided by angles, this is because when encoding an angle, a message from a 2-hop neighbor to a 1-hop neighbor is involved (e.g. E_sk in the interaction block in SphereNet). Since we have O(N^2) angles (where N is the average number of neighbors) defined around a central node, there are also O(N^2) messages involved when updating one node embedding in each hidden layer. However, if we don't use a message from a 2-hop neighbor to a 1-hop neighbor when encoding an angle, the computations can be much more efficient.
> > > > > > > >
> > > > > > > > Without using E_sk between 2-hop neighborhoods and 1-hop neighborhoods, we can find a way to define and use fixed descriptors or fingerprints (e.g. inspired by traditional fingerprints that encode a molecule) to encode the related 3D information with related atom features. Note that each resulting fixed descriptor D encodes the features for completeness related to a 1-hop neighbor (e.g. s_k in Figure 2 of SphereNet paper), so that D can replace the computations associated with E_sk in the interaction block. D can be transformed with a non-linear function to be added to e_k, which is the message for 1-hop message passing. The remaining parts of SphereNet can be unchanged. By using fixed descriptors rather than learned embeddings to encode the 3D information involved in the original second step message passing, it is possible to only use 1-hop message passing to achieve completeness with O(N) complexity.

---

> > > > > > > > > ### Author Response · Authors · 2021-11-29
> > > > > > > > > **Thank you for your idea on efficiency**
> > > > > > > > >
> > > > > > > > > Dear reviewer epnb,
> > > > > > > > >
> > > > > > > > > Thanks for your suggestion and for proposing this interesting idea.
> > > > > > > > >
> > > > > > > > > Basically, you suggest removing SMP and all the 3D information (distance, angle, torsion) in the learning process, and using fingerprints D instead. D could be a hash set of 3D subgraphs whose structures are fixed. Then the similar purpose of SMP can somehow be reached by a combination of subgraphs in D.
> > > > > > > > >
> > > > > > > > > The fingerprint is an effective technique for 2D graphs [1]. If we only care about 2D connections, the hash is finite and the whole graph is easy to get by combining multiple subgraphs.
> > > > > > > > >
> > > > > > > > > However, the fingerprint for 3D structures is tricky. The position of an atom in 3D space is continuous instead of discrete. Hence, search space is infinite, so is the hash D.
> > > > > > > > >
> > > > > > > > > Your inhere idea reminds me of a paper [2] you suggested previously. We can perform discretization in spatial space as [2] does, then the position space is finite, making the process feasible and valid. However, completeness cannot be achieved due to the discretization in spatial space. For sure, this is an interesting idea, which should be able to solve a lot of problems where completeness is not a motivation.
> > > > > > > > >
> > > > > > > > > Overall, your idea combining the work [2] should lead to an interesting and valid method, slightly different from SMP though on motivation and novelty. We appreciate so much you making efforts to help improve the efficiency.
> > > > > > > > >
> > > > > > > > > Seems previous concerns are addressed. The discussion here is for a super interesting, valid but different approach. Could you raise your score from 5.5 to the acceptance range? Thank you!
> > > > > > > > >
> > > > > > > > > > Reference
> > > > > > > > >
> > > > > > > > > [1] Convolutional Networks on Graphs for Learning Molecular Fingerprints
> > > > > > > > >
> > > > > > > > > [2] Spherical kernel for efficient graph convolution on 3d point clouds

---

> > > > > ### Author Response · Authors · 2021-11-28
> > > > > **Looking forward to your reply**
> > > > >
> > > > > Dear reviewer epnb,
> > > > >
> > > > > We think we gradually addressed your updated concerns along the way, and have made novelty/motivation/main contributions clear. For the remaining questions, we think they are properly answered in our last author responses. As the open discussion period will end soon, could you please check and let us know? Thanks!
> > > > >
> > > > > Motivation and novelty were your main concerns from the beginning, and you now agree with them. We do believe based on how we have been making efforts to address your concerns, our detailed discussions, and its clear motivation/technical novelty/helpful empirical studies, the paper should meet the acceptance bar. Could you please consider raising the rating score if you think your main concerns have been addressed? Thank you for your time!

---

> ### Comment · Area_Chair_Jtae · 2021-11-18
> **Comments on author feedback and other reviews**
>
> This review seems to be the most negative ones of the 3 available. Could you please look at the author feedback and the other reviews and say how that changes your initial review?

---

### Decision · Program_Chairs · 2022-01-20

**Decision:**

Accept (Poster)

**Comment:**

The paper considers representation learning of 3D molecular graphs.
The authors propose a message passing scheme using spherical coordinates. It is
tested on three datasets of 3D moleclular graphs. The authors offer an in depth
analysis of different aspects, with an extensive experimentation of the method.

Strengths:

- The SMP introduces an interesting method to alleviate the computation cost issue in SCS from O(nk^3) to O(nk^2). This method is important and can be generalized to more broad types of tasks.
- This is an empirical work, and the experimental results support the effectiveness of SMP.
- The proposed MP approach can better distinguish certain structures than some existing models.
- Incorporating torsion information when representing 3D molecules is novel and helpful
- While message passing methods on graphs exploit only the connectivity, this work shows an interesting method to include the embedding information in the case of geometrical graphs.

Weaknesses:

- The proposed SMP scheme in Eq. (1) lacks novelty since it basically enriches the GN framework in [1] with geometry features
- the architecture of the proposed SphereNet is similar to DimeNet
- Why SMP is better than Cartesian coordinate system (CCS) is not well explained.

Overall, a majority of reviewers are in favor of acceptance and a third reviewer is happy with either acceptance or rejection and does not give strong reasons for rejecting the paper. My recommendation is, therefore, acceptance. I recommend the authors use the reviewers comments to improve the paper for its camera-ready version.